# Molecular Mechanisms and Regulation of Mammalian Mitophagy

**DOI:** 10.3390/cells11010038

**Published:** 2021-12-23

**Authors:** Vinay Choubey, Akbar Zeb, Allen Kaasik

**Affiliations:** Department of Pharmacology, Institute of Biomedicine and Translational Medicine, University of Tartu, Ravila 19, 50411 Tartu, Estonia; akbar@ut.ee (A.Z.); allen.kaasik@ut.ee (A.K.)

**Keywords:** PINK1, PARKIN, mitophagy, autophagy, FUNDC1, BNIP3, cardiolipin, Parkinson’s disease, quality control

## Abstract

Mitochondria in the cell are the center for energy production, essential biomolecule synthesis, and cell fate determination. Moreover, the mitochondrial functional versatility enables cells to adapt to the changes in cellular environment and various stresses. In the process of discharging its cellular duties, mitochondria face multiple types of challenges, such as oxidative stress, protein-related challenges (import, folding, and degradation) and mitochondrial DNA damage. They mitigate all these challenges with robust quality control mechanisms which include antioxidant defenses, proteostasis systems (chaperones and proteases) and mitochondrial biogenesis. Failure of these quality control mechanisms leaves mitochondria as terminally damaged, which then have to be promptly cleared from the cells before they become a threat to cell survival. Such damaged mitochondria are degraded by a selective form of autophagy called mitophagy. Rigorous research in the field has identified multiple types of mitophagy processes based on targeting signals on damaged or superfluous mitochondria. In this review, we provide an in-depth overview of mammalian mitophagy and its importance in human health and diseases. We also attempted to highlight the future area of investigation in the field of mitophagy.

## 1. Introduction

Mitochondria in the cell are double-membraned organelles, which hold a central role in energy production [1], essential biomolecule synthesis [2], calcium buffering [3] and importantly, in pro-survival or pro-apoptotic signaling [4]. These functions are carried out by over 1200 proteins [5], although the exact composition of mitochondrial proteins can vary greatly depending on the cell stage, type and environment [5]. Due to the functional importance of mitochondria, any disturbance in this proteome will leave a profound impact on cell fate and could result in diseases ranging from neurodegenerative disorders and heart diseases to diabetes and cancer [6,7,8,9,10,11]. 

Owing to their role in energy production by the electron transport chain, mitochondria are the primary site of reactive oxygen species (ROS) generation [8]. Excessive ROS production damages mitochondria further, creating a vicious circle [12]. Although ROS are important signaling molecules [13], they are detrimental to the cell when in excess [11,12,14]. To control the ROS accumulation, mitochondria are equipped with different types of antioxidant systems, such as mitochondrial superoxide dismutases (SOD), thioredoxin reductase and glutathione peroxidase [15]. The second potential threat to mitochondrial health arises from mitochondrial plasticity, which demands a constant change in the mitochondrial proteome to adapt to cellular needs [16,17]. Under such fluctuating conditions, the mitochondrial proteostasis is maintained by a robust mitochondrial import system collaborating with mitochondrial proteases and chaperones [5,16,18,19,20]. This system not only delivers the functional proteins to mitochondria but also prevents the accumulation of non-functional or non-desirable proteins under given conditions. Besides, the ROS defense and proteostasis system, mitochondrial dynamics is also involved in the mitochondrial quality control, as slightly damaged mitochondria may fuse with healthy ones or may separate from the damaged mitochondrial part via fission [21,22]. These multilayered mitochondrial quality control systems, including ROS defense, proteostasis and mitochondrial dynamics, all work in concert to preserve mitochondrial function and normal cell physiology. 

However, if the damage to mitochondria overwhelms the capacity of the quality control systems, then these mitochondria should be removed from mitochondrial network by mitophagy, the selective degradation of mitochondria in lysosomes by autophagy (Figure 1) [23,24].

Lysosomal mitochondria were first observed in mammalian cells by early electron microscopy studies by De Duve and Wattiaux in 1966 [25] but this was first referred to as selective mitochondrial autophagy or mitophagy by J. J. Lemasters in 2005 [26]. Mitophagy removes old, superfluous or dysfunctional mitochondria before they become a threat for cells. Since that initial observation, extensive studies have identified different mechanisms of mitophagy activated under different kinds of stresses, such as oxidative damage, hypoxia, mitochondrial depolarization and mitochondrial DNA (mtDNA) damage [23,24,27,28,29,30,31,32,33,34]. Based on the targeting signals on damaged or superfluous mitochondria that initiate mitophagy, this process can be divided into:➢Ubiquitin-dependent mitophagy
PARKIN dependent (PINK1-PARKIN pathway)PARKIN independent but ubiquitin dependent mitophagy:➢Ubiquitin-independent or receptor based mitophagy
Apoptosis related proteins as mitophagy receptors or inhibitorOther mitophagy receptors➢Lipid based mitophagy
Cardiolipin basedSphingolipid Based➢Micromitophagy

## 2. Ubiquitin-Dependent Mitophagy

Ubiquitin-dependent mitophagy relies on ubiquitin as a signal on the surface of damaged or superfluous mitochondria. Ubiquitin-marked mitochondria will then recruit autophagic machinery that leads to their degradation by mitophagy. Ubiquitination of mitochondrial proteins is achieved by different pathways that lead to the initiation of mitophagy, but below are described only the most well-studied ones.

### 2.1. PARKIN Dependent (PINK1-PARKIN Pathway)

The landmark studies from Richard Youle’s lab on the PINK1-PARKIN pathway revolutionized the field of mitophagy [35,36], which made this pathway the best-studied mitophagy pathway among all others. This pathway primarily depends on the mitochondrial serine/threonine protein kinase PINK1 and the cytosolic E3 ubiquitin ligase PARKIN [29,37]. Earlier studies in *Drosophila* found that PARKIN and PINK1 were essential for mitochondrial function and work in the same pathway [38,39]. Later studies established the roles of PINK1 and PARKIN in mitochondrial biology more precisely [29,37,40]. PINK1 and PARKIN were found to monitor diverse aspects of mitochondrial health, ranging from mitochondrial quality control and mitochondrial dynamics to mitochondrial biogenesis [29]. Thus, not surprisingly, mutations in *PINK1* and *PARKIN* genes (resulting in mitochondrial dysfunction) are implicated in several neurodegenerative diseases such as Parkinson’s disease (PD), Alzheimer’s disease (AD) and Multiple Lateral Sclerosis (MLS) [29,41] 

PINK1 utilizes the canonical presequence-driven mitochondrial import pathway to monitor mitochondrial health [29,37,42]. Under basal conditions, PINK1 is imported into the polarized mitochondria through the mitochondrial translocases of the outer and inner membranes (TOM and TIM), with the help of its positively charged amino-terminal mitochondrial targeting sequence [37,42] (Figure 2). Following the import of PINK1, it is cleaved twice when it is in the inner mitochondrial membrane. The first cleavage by the matrix processing peptidase (MPP) removes the mitochondrial targeting sequence, while the second cleavage occurs in the transmembrane domain between Ala103 and Phe104 by the inner membrane protease Presenilin-Associated Rhomboid-Like protein (PARL) [37,42] (Figure 2). As PARL is involved in the cleavage of PINK1 and PGAM5, it is now called the PINK1/PGAM5-associated rhomboid-like protease [42,43,44]. 

Coming back to PINK1 import, PARL-cleavaged PINK1 is released into the cytoplasm and degraded rapidly by the proteasome via the N-end rule ubiquitination machinery, keeping the basal levels of PINK1 low [45]. However, it has been suggested that N-end rule might not be the primary mechanism of PINK1 degradation. It might also be degraded through a proteasome-dependent mechanism relying on the polyubiquitination of the mature 52-kDa form of PINK1 preferential at K137. This hypothesis is supported by evidence that the bulk of ubiquitinated (Ub)-PINK1 is mitochondrially anchored rather than cytosolic. Secondly, the N-terminal phenylalanine (F104) of PINK1 was not detected in the cytosol but in the mitochondrial outer membrane (MOM) where it would be inaccessible to the cytosolically localized N-end rule ubiquitination machinery [46]. Later, the same group proposed that 52 kDa PINK1 localizes at the mitochondrial–endoplasmic reticulum (ER) interface, where components of ER-associated degradation pathway, such as the E3 ligases GP78 and HRD1, catalyze PINK1 ubiquitination and promote its proteasomal degradation to maintain PINK1 content low in healthy mitochondria [47]. Contrary to the PINK1 degradation mechanism, it is largely accepted that the full-length PINK1 is stabilized on the MOM upon mitochondrial membrane potential dissipation [36], excessive mitochondrial reactive oxygen species (ROS) generation [48,49] and/or mitochondrial protein aggregation [50,51]. The mitochondrial depolarization or bioenergetic deficit leads to the inhibition of the TIM23-mediated import of PINK1 via indirect modulation by the adenine nucleotide translocator (ANT) [52]. In another study, PINK1 import arrest under mitochondrial depolarization was not found solely dependent on Tim23 inactivation but also by an actively regulated “tug of war” between Tom7 and OMA1 [53]. Irrespective of the mechanism of PINK1 import arrest, it is generally accepted that PINK1 forms a 700 kDa complex with a translocase of the outer membrane on depolarized mitochondria after its import is arrested [54] (Figure 2). In this MOM complex, PINK1 molecules likely exist as dimers, however, the role of dimeric PINK1 in its kinase activity remains to be established. PINK1 dimerization is proposed to facilitate PINK1’s autophosphorylation on Ser228 and Ser402 residues in its kinase domain [55,56]. 

After stabilization on MOM and autophosphorylation, PINK1 affects PARKIN in two ways. First, it phosphorylates pre-existing ubiquitins already conjugated to MOM proteins at Ser65. PARKIN’s high affinity for phosphorylated ubiquitin (pSer65–Ub) drives its translocation from the cytosol to mitochondria [57,58] as well as partially activates its ubiquitination activity [59]. This partially activated pSer65–Ub-bound PARKIN, ubiquitinates MOM proteins further, providing more substrate for PINK1. This generates more pSer65–Ub which attracts a second wave of PARKIN from the cytosol to mitochondria (Figure 2). PINK1 also phosphorylates pSer65–Ub-bound PARKIN at Ser65 in the ubiquitin-like domain, activating its autoinhibited E3 ubiquitin ligase activity nearly 4400-fold [60,61,62], which further drives ubiquitin chain formation on the MOM proteins. The binding of pSer65-Ub to pSer65-PARKIN is 19 times stronger than to unphosphorylated PARKIN, thus further favoring the retention of fully active pSer65-PARKIN on damaged mitochondria [63]. However, there appears no particular order to it. PINK1 could phosphorylate first PARKIN, since PARKIN can be phosphorylated and activated by PINK1 without its initial encounter with pSer65-Ub; additionally, ubiquitin could be phosphorylated independently of PARKIN. In both scenarios, activated PARKIN conjugates further ubiquitins to MOM proteins, which are then phosphorylated by PINK1. This forms a positive feedback loop that amplifies the initial signal, resulting in extensive PARKIN recruitment and ubiquitination [57,59,62] (Figure 2). Interestingly, PINK1 kinase activity is sufficient for PARKIN recruitment, as the forced expression of PINK1 on peroxisomes or lysosomes leads to PARKIN translocation to respective organelles [54].

The recruitment of PARKIN to mitochondria leads to the ubiquitination and further proteasomal degradation of multiple MOM proteins [64] such as Mfn1/2 [65,66,67], Miro1/2 [64,68], VDAC [69,70], TOMs [71] and mitochondrial hexokinase [64]. The extraction of ubiquitinated protein for degradation from MOM is accomplished by p97, an AAA+ ATPase which accumulates in mitochondria along with the proteasome in a PARKIN-dependent manner [66,72]. Another possible way of proteasome accumulation onto mitochondria is the direct interaction between PARKIN’s ubiquitin-like domain and the Rpn13 subunit of the 26S proteasome [73]. This interaction attracts the proteasome to mitochondria and facilitates the proteasomal degradation of certain MOM proteins and PARKIN itself [73]. Ubiquitination and proteasomal degradation possibly occur in a bi-phasic manner [74]. The primary targets of the first phase are MOM proteins, which lead to the rupture of MOM [71]. Rupture of MOM opens the door for the second phase of ubiquitination of proteins localized inside mitochondria (Figure 3) [71,74]. A total of 36 MOM substrates of PARKIN have been identified with high confidence [64]. Interestingly, some mitochondrial substrates were earlier considered not only as substrates but also as PARKIN receptors, but it was later suggested that phosphorylated ubiquitin is the main receptor for PARKIN at mitochondria [58]. Nevertheless, we recently demonstrated that Miro proteins are not only the substrates for PINK1/PARKIN-dependent degradation, but they can also function as a calcium-dependent docking site and safety switch for PARKIN recruitment [75]. 

It is relevant to note that translocated PARKIN forms ubiquitin chains on MOM proteins with linkage types that are characteristic of both autophagy and proteasomal degradation [76,77,78]. PARKIN has been shown to a the Lys48-linked polyubiquitin chains onto several MOM proteins [78] and typically, Lys48 polyubiquitin chains consist of a minimum of four ubiquitin moieties target proteins for degradation via the proteasome [79,80]. Fast proteasomal removal of mitofusins prevents damaged mitochondria from fusing with the healthy mitochondrial network and segregates them for mitophagy degradation [78]. Similarly, Miro proteins involved in mitochondrial motility, are removed by the proteasomal degradation, leading to mitochondrial movement arrest to facilitate mitophagy [81]. However, whether these events are really prerequisites for mitophagy is currently uncertain and demands further investigation.

In addition to Lys48 polyubiquitin chains, PARKIN also as polyubiquitin chains that are linked through Lys63, Lys6 or Lys11 to MOM proteins [62]. The overall abundance of Lys6 linked polyubiquitin chains does not increase with proteasome inhibition, suggesting that Lys6 chains do not lead to proteasomal degradation [82]. However, the functional relevance of different types of polyubiquitin chains in mitophagy is so far not conclusively established. 

PARKIN-mediated ubiquitination can also be affected by the deubiquitinating enzymes (DUBs) (Figure 3A). In fact, mitochondrially localized ubiquitin carboxyl-terminal hydrolase 30 (USP30) deubiquitinase was found to reverse the PARKIN-dependent ubiquitination of TOMM20 and Miro1 [83]. Thus, the deubiquitinating enzymes oppose PINK1-PARKIN mitophagy [83] and it has been suggested that this might prevent the degradation of healthy mitochondria [84]. However, the polyubiquitination and deubiquitination process is a wasteful process and cells should know how to avoid that and proceed with mitophagy. The answer is provided by an extensive study demonstrating that Ser65 phosphorylated ubiquitin dimers are particularly resistant for the cleavage by 31 different DUBs [85]. This suggests that at mitochondria, PINK1-dependent phosphocapping of ubiquitin is making ubiquitinated MOM proteins DUB-resistant. An alternative explanation would be that PARKIN activation outpaces the ubiquitin chain removal to an extent that would overcome deubiquitinase (USP30)-mediated antagonization [84]. 

As mentioned above, PARKIN-ubiquitinated MOM proteins could undergo either proteasome mediated degradation (Figure 3B) or disassembly of ubiquitin chains by DUBs (Figure 3A). As the third possibility, the PARKIN-ubiquitinated MOM proteins can also trigger the recruitment of the autophagy cargo receptors to mitochondria (Figure 3C). The autophagy receptors bind at one end to the ubiquitinated cargo (via their ubiquitin binding domains, UBD) and at other end (via their LC3-interacting region, LIR) to Microtubule Associated Protein-Light Chain 3 (MAP-LC3 or just LC3) that is localized on the autophagosomal membranes [86]. Thus, autophagy receptors facilitate the delivery of ubiquitinated cargo to the autophagosome for autophagic degradation [86]. In mammalian cells, five autophagy receptors have been linked to ubiquitin dependent mitophagy: p62, AMBRA1, Nuclear Domain 10 Protein 52 (NDP52), Optineurin (OPTN) and TAX1BP1 [87]. However, p62 was found to be required for the perinuclear clustering of depolarized mitochondria, but not for mitophagy [88]. More importantly a knock-out study of five autophagy receptors (p62, Neighbor of *BRCA1* gene 1 (NBR1), NDP52, OPTN and TAX1BP1) revealed that NDP52 and OPTN could rescue mitophagy redundantly in these Penta KO HeLa cells [89]. The study additionally showed that the compensatory ability of OPTN for NDP52 during mitophagy required TANK binding Kinase 1 (TBK1) [89]. TBK1 was shown to phosphorylate OPTN at multiple sites in a PINK1- and PARKIN-dependent manner, leading to the increased affinity of OPTN for K63-Ub chains and ATG8 (yeast equivalent of LC3) proteins [90,91]. Importantly, mutations in *TBK1* and *OPTN* have been genetically linked with amyotrophic lateral sclerosis. These mutations in *TBK1* and *OPTN* often disrupt their respective proteins association suggesting the significance of their association in removing autophagy cargo [92,93,94]. Moreover, the autophagy cargo receptors, i.e., OPTN and NDP52, promote the biogenesis of phagophores in close proximity to mitochondria by recruiting the autophagy-initiating factors ULK1 (unc-51-like autophagy activating kinase 1), DFCP1 (double FYVE-domain containing protein 1) and WIPI1 (WD repeat domain, phosphoinositide interacting 1) upstream to LC3 recruitment [89]. After the engulfment of targeted mitochondria by the autophagosome, it fuses with acidic hydrolases containing lysosomes for the complete degradation of mitochondria [95]. 

Interestingly, PINK1 and PARKIN play not only key roles in mitophagy degradation, but also regulate mitochondrial biogenesis (discussed later in the review). 

Though the PINK1–PARKIN mitophagy pathway is the most studied mitophagy pathway, it has been often criticized for the mostly studied in in vitro with PARKIN overexpression in non-neuronal immortalized cells in the presence of mitochondrial toxins, or in conditions that are far from physiological [96,97,98,99]. Moreover, recent in vivo studies indicate that PINK1 and PARKIN are not critical for basal mitophagy in various tissues, including the brain [100,101]. Therefore, recent studies have been focused on the identification of the alternative mitophagy pathways.

In that line, several pathways not following the classical PINK1–PARKIN pathway started to emerge, such as pathways independent of PARKIN, where other Ub ligases prime the mitochondria for mitophagy. Therefore, we next focus on PARKIN-independent, but ubiquitin-dependent, mitophagy. 

### 2.2. PARKIN-Independent but Ubiquitin-Dependent Mitophagy

In recent years, several studies have identified mitophagy pathways that are ubiquitin-dependent, but independent of PINK1 and/or PARKIN (Figure 4). These pathways are suggested to act either in parallel or in addition to the classical PINK1–PARKIN pathway. Their molecular mechanisms, leading to the ubiquitination of mitochondrial proteins and, subsequently, mitophagy, have started to emerge. 

#### 2.2.1. Glycoprotein 78 Mitophagy

Glycoprotein 78 (GP78) is an endoplasmic reticulum (ER) membrane–anchored ubiquitin ligase (E3) that is a key component of the ER-associated degradation (ERAD) and is found localized to the mitochondria-associated ER domain [102]. Over-expression of GP78 has been found to ubiquitinate Mfn1 and Mfn2, inducing their proteasomal degradation, leading to mitochondrial fragmentation [103]. Moreover, over-expressed GP78 was found to induce mitophagy upon mitochondrial depolarization in COS-7 cell lines by recruiting LC3 to the GP78-positive ER domains, closely associated with depolarized mitochondria. The GP78 induced mitophagy was dependent on GP78’s ubiquitin ligase activity, and the autophagy protein Atg5 and Mfn1, but it was PARKIN-independent since it occurred in *PARKIN*-null HeLa cells, as well as in *PARKIN* knockdown HEK293 cells [103]. Interestingly, GP78 activity is regulated by the cytosolic E3 ubiquitin ligase mahogunin RING finger 1 (MGRN1), which ubiquitinates GP78 in trans through noncanonical K11 linkages, promoting the degradation of GP78 [102]. This maintains constitutively low levels of GP78 in healthy cells and downregulates mitophagy. Whereas, mitochondrial stresses by CCCP or higher cytosolic Ca^2+^ perturb the interaction between MGRN1 and GP78, leading to GP78 accumulation and triggering PARKIN-independent mitophagy [104] (Figure 4A). Interestingly, ERAD machinery containing GP78 has been proposed to regulate PINK1 levels in human and monkey cell lines also [47]. 

#### 2.2.2. PINK1-SYNPHILIN1-SIAH1 Mitophagy

Another reported ubiquitin-dependent but PARKIN-independent mitophagy pathway involves the PINK1, SYNPHILIN1 and seven in absentia homolog 1 (SIAH1) [105]. In this pathway, PINK1 recruits SYNPHILIN1 to the mitochondria, as SYNPHILIN1 can interact with the full-length and cleaved form of PINK1 in rat brain tissues and cultured cells [105]. Though SYNPHILIN1 was observed to preferentially interact with cleaved PINK1, the authors suggested that this could be due to the nature of the antibody used [105]. The PINK1-mediated localization of SYNPHILIN1 to mitochondria causes depolarization of the mitochondria that leads to stabilization of uncleaved PINK1 at the organelle. This further promotes translocation of SYNPHILIN1 to the mitochondria, which in turn recruits the E3 ubiquitin ligase SIAH1 to the mitochondria as SYNPHILIN1 has the ability to interact with SIAH1 also. The SIAH1 subsequently ubiquitinates mitochondrial proteins that results in the recruitment of the autophagosome marker LC3 and the lysosome marker Lamp1 to the mitochondria for mitophagy [105] (Figure 4B). This PINK1–SYNPHILIN1–SIAH1 induced mitophagy did not depend on an exogenous depolarizing agent or on PINK1-mediated phosphorylation of SYNPHILIN1or ubiquitin, as well as did not involve PARKIN. Although this pathway was independent of PINK1 kinase activity, PD-related PINK1 mutants (G309D, A168P and L347P) decreased this mitophagy by more than 50%, pointing that not only the PINK1–PARKIN pathway but alternative mitophagy pathways could also be affected by *PINK1* mutations [105]. Since this pathway is independent of PINK1 kinase activity, it may represent a possible new drug target in disease cases involving PINK1 kinase domain mutants. Besides SIAH1, the other members of the SIAH family have been found to regulate different aspects of mitochondrial biology. For example, SIAH2 has been shown to regulate mitochondrial dynamics by controlling the degradation of Mfn1 and Drp1 in cortical neurons under hypoxia [106]. Moreover, SIAH2 negatively regulates mitochondrial biogenesis by ubiquitination and subsequent proteasomal degradation of nuclear respiratory factor 1 (NRF1), a crucial factor for mitochondrial biogenesis [107] under hypoxic microenvironments [108]. Whereas, PARKIN, another E3 ubiquitin ligase, promotes mitochondrial biogenesis by degradation of the repressor of PGC-1α, PARIS [109,110].

#### 2.2.3. MUL1-Based Mitophagy

The mitochondrial E3 ubiquitin ligase 1 (MUL1), also known as mitochondrial-anchored protein ligase (MAPL) [111] or mitochondrial ubiquitin ligase activator of NF-κB (MULAN) [112], has been reported to be involved in mitophagy induction [113,114,115]. However, the mechanisms proposed for MUL1-mediated mitophagy are poorly understood and lack consensus (Figure 4C). 

MUL1 is a MOM-embedded protein, with its RING finger facing the cytoplasm and its intermembrane domain located in the intermembrane space (IMS). MUL1 is a multifunctional protein but its major biological functions are ubiquitination and sumoylation. Like other ubiquitin ligases, MUL1 should interact with E2-conjugating enzymes for ubiquitination. Currently, four E2-conjugating enzymes (Ube2E2, Ube2E3, Ube2G2 and Ube2L3) are identified as specific interactors of MUL1 [114]. Among them, the Ube2E3 was implicated in the induction of mitophagy in HEK293 cells treated with CCCP. In this case, MUL1 was found to bind GABARAP (GABA receptor-associated protein), a member of the Atg8 family that plays a major role in mitophagy [114]. The binding with GABARAP requires an LC3-interacting region (LIR), located in the RING finger domain of MUL1, as well as the presence of Ube2E3, suggesting a plausible mechanism of MUL1-induced mitophagy [114]. 

Another study demonstrated a role of MUL1 in selenite-induced mitophagy, which was ULK1- and Atg5-dependent but was PARKIN-independent [115]. The study proposed that under normal conditions, MUL1 monitors the MOM quality and prevents ULK1 from initiating mitophagy. However, under stress conditions or a higher intermembrane spatial ROS, ULK1 translocates to the mitochondria to initiate mitophagy. Similarly, selenite promoted the partial translocation of ULK1 to mitochondria, where it interacted with MUL1, which, in turn, ubiquitinated ULK1 for proteasomal degradation, making ULK1 a novel substrate of MUL1 [115]. However, the mechanism proposed was unclear, as MUL1-enhanced mitophagy paradoxically promoted ULK1 degradation. Additionally, two highly conserved cysteine residues in MUL1 were proposed to play an important role in MUL1-induced mitophagy by ROS and selenite, since treatment with anti-oxidants prevented the mitophagy induction [115].

As an E3 ubiquitin ligase, MUL1 ubiquitinates many functional and signaling proteins, such as mitofusin2 (Mfn2), Akt, p53 and ULK1, leading to mostly their degradation [116]. On the other hand, MUL1 promotes the sumoylation of DNM1L/Drp1 which stabilizes it and leads to mitochondrial fission [116]. This, together with the reduction in mitochondrial fusion via ubiquitination and degradation of Mfn2, results in an overall fragmented mitochondrial morphology [112], creating an environment favoring mitophagy. This ability of MUL1 to regulate Mfn2 has been proposed as a mitophagy-inducing mechanism in Omi/HtrA2^(−/−)^ mouse embryonic fibroblasts (MEFs) treated with CCCP [113]. The study also suggested Omi/HtrA2 protease as a negative regulator of MUL1, since it accumulated in Omi/HtrA2^(−/−)^ MEFs and in different tissues of motor neuron degeneration-2 (mnd2) mutant mice [113]. Moreover, MUL1-mediated Mfn2 degradation was attributed to the suppression of PINK1 or PARKIN mutant phenotypes in *Drosophila* and mouse neurons. In contrast, double mutants of *MUL1,* with either *PINK1* or *PARKIN,* aggravates severe phenotypes [117]. The study suggested that MUL1 functions in a pathway parallel to the PINK1–PARKIN pathway and could compensate for the loss of *PINK1* or *PARKIN* in *Drosophila* and mammals [117].

Another recent study supported the mitophagy-inducing role of MUL1 by demonstrating the stabilization of PINK1 and its subsequent mitophagy in mammalian cells treated with the anticancer drug gemcitabine [118]. This mitophagy did not require mitochondrial depolarization and took place even in *PARKIN*-deficient HeLa cells [118]. Interestingly, stabilization of PINK1 in this study required MUL1 but the mitophagy mechanism was again unclear [118].

Contrary to the abovementioned studies, one recent report observed that MUL1 resists PARKIN translocation, as well as mitophagy, in response to mild/chronic mitochondrial stress [119]. This resistance was observed under early stress conditions to allow the recovery of stressed mitochondria instead of their mitophagy in non-dividing and post-mitotic neurons [119]. The study proposed MUL1 with Mfn2 forms a checkpoint that maintains the integrity of neuronal mitochondrial morphology and interplay between mitochondria and endoplasmic reticulum (ER). MUL1-deficient neurons trigger a biphasic mitochondrial response to mild stress. In the first phase, stabilized Mfn2 leads to transient hyper-perfusion and also antagonizes Mito–ER contacts. In the later phase, the disturbance of communication between Mito–ER increases intracellular Ca^2+^, which leads to the activation of calcineurin, Drp1 and PARKIN-mediated mitophagy. In contrast to deficiency, overexpression of MUL1 suppresses PARKIN translocation and mitophagy [119]. 

#### 2.2.4. SQSTM1 Based Ubiquitin Based Mitophagy

The autophagy adapter p62/SQSTM1 was earlier demonstrated to connect mitochondria to autophagosomes by binding to polyubiquitinated MOM proteins and LC3 simultaneously [69,120]. However, later studies suggested that p62 is required only for the perinuclear clustering of depolarized mitochondria, but not for mitophagy [88]. In recent times, new roles of p62/SQSTM1 have started to emerge in the mitophagy pathway. 

To that end, a Keap1-Nrf2 (kelch-like ECH-associated protein 1-Nuclear factor-erythroid factor 2-related factor 2) PPI inhibitor HB229 (PMI) was shown to activate p62/SQSTM1 dependent mitophagy by up-regulating *p62/SQSTM1* gene expression via Nrf2 activation [121,122]. The upregulated p62/SQSTM1 translocated to the mitochondria and enhanced polyubiquitination of the MOM proteins that, in turn, recruited LC3 for mitophagy [122]. Interestingly, PMI was found not to cause depolarization or damage to mitochondria; instead, it was found to increase mitochondrial superoxide production. This superoxide production was suggested to play an important role in PMI induced mitophagy [122,123] (Figure 4D). Moreover, PMI induced mitophagy was observed, even in the cells lacking a fully functional PINK1–PARKIN pathway, but not in Nrf2^−/−^ and p62/SQSTM1^−/−^ MEFs [122]. The PMI was also not found to alter the expression of other mitophagy receptors in SH-SY5Y cells [123]. However, the study did not reveal the specific E3 ubiquitin ligase involved in the polyubiquitination of the MOM proteins after p62/SQSTM1 mitochondrial translocation. 

A possible link between p62/SQSTM1 and polyubiquitination of the MOM proteins came from an interesting report suggesting that during mitophagy, p62/SQSTM1 can itself regulate the polyubiquitination of mitochondrial proteins via Keap1 and its associated proteins [124]. In this study, mitophagy intermediates were studied from multiple organs of dynamin-related GTPase (mediates mitochondrial division) *Dnm1l/Drp1* KO mice [124]. In these organs (brain, heart and liver), the loss of *Dnm1l/Drp1* enlarged mitochondria and halted mitophagy with mitochondria having accumulated p62/SQSTM1, ubiquitinated proteins and LC3 on their surfaces. In contrast, an additional KO of *p62/Sqstm1* dramatically decreased the mitochondrial ubiquitination in *dnm1l* KO hepatocytes, suggesting p62/SQSTM1 was responsible for polyubiquitination of the MOM proteins [124]. The p62/SQSTM1 was proposed to influence ubiquitination, by associating with Keap1 along with its companions cullin-RING ubiquitin ligase with the E3 ligase RBX1 [124]. Both KEAP1 and RBX1 are recruited to the mitophagy intermediates by p62/SQSTM1 in *dnm1l* KO hepatocytes, and knockdown of *Rbx1* also decreased mitochondrial ubiquitination (Figure 4D). Moreover, their data suggested that recruitment of p62/SQSTM1 to mitochondria and the polyubiquitination of mitochondrial proteins were independent of PINK1 as well as PARKIN, as their KO did not affect that. Unlike PMI studies, in this case p62/SQSTM1-driven mitophagy was independent of the Nrf2 (NFE2L2) pathway [121,122,124]. 

In contrast to the abovementioned studies, there are reports suggesting that p62/SQSTM1 is dispensable for mitophagy. For example, CCCP or oligomycin and antimycin-induced mitophagy was found to be unaffected by loss of *p62/SQSTM1* in neuroepithelial stem cells (NESC) and in differentiated neurons derived from reprogrammed fibroblasts, obtained from patients carrying nonsense mutations at the 5′ end of *p62/SQSTM1* [125]. Similarly, another study suggested p62/SQSTM1 was not essential for mitophagy in iPSC-derived *p62/SQSTM1-KO* neurons [126]. In addition, as mentioned earlier, p62/SQSTM1 was found to be required only for the perinuclear clustering of depolarized mitochondria, but not for mitophagy [88].

However, further studies on p62/SQSTM1 are very important with respect to human health since deficiency or loss of *p62/SQSTM1* could lead to amyotrophic lateral sclerosis [127,128,129], frontotemporal dementia [130] and childhood- or adolescence-onset neurodegenerative disorders [131,132]. Very importantly, the pathogenic mechanism that contributes to p62/SQSTM1-related neurodegeneration remains poorly understood.

## 3. Ubiquitin-Independent or Receptor-Based Mitophagy Pathways

In addition to Ub-driven mitophagy, several mitophagy mechanisms have now been reported which target mitochondria to autophagosomes independently of mitochondrial ubiquitination. This kind of mitophagy depends on autophagy receptors which are capable to interact directly with LC3 and/or GABARAP (GABA-receptor-associated protein) through typical or atypical LC3 Interacting Region (LIR) motifs such as BNIP3 (BCL2/adenovirus E1B 19 kDa protein interacting protein 3) [133], NIX (NIP3-like protein X)/BNIP3L [134], FUNDC1 (FUN14 domain-containing 1) [135], BCL2L13 (BCL2 like 13) [136] and FKBP8 (FKBP prolyl isomerase 8) [137] in mammals. These receptors are mostly located on the MOM and depend on their LIR for mitochondrial clearance. The receptor-mediated mitophagy components are regulated by transcriptional or post-transcriptional modification in response to different mitochondrial stresses. It is proposed that receptor-mediated mitophagy operates at high rate under basal level or chronic stress situations, whereas the PINK1–PARKIN pathway compensates for acute, chemical-insult-mediated mitochondrial dysfunction [95,101].

### 3.1. Apoptosis Related Proteins as Mitophagy Receptors or Inhibitors

The BCL-2 family proteins are well-known for their major regulatory role in apoptosis [138], but in recent times their key role in mitophagy pathways have also been recognized [139]. Ubiquitination-independent mitophagy in several instances was found to be regulated either by pro-apoptotic proteins such as BNIP3, BNIP3L and BCL2L13, belonging to the BCL2 family [133,134,136], or by an anti-apoptotic protein, FK506 binding protein 8 (FKBP8), belonging to FKBP family [137,140].

#### 3.1.1. BNIP3 and Nix/BNIP3L in Mitophagy 

BCL2/adenovirus E1B 19 kDa protein-interacting protein 3 (BNIP3) and BNIP3-like (BNIP3L) or Nip3-like protein X (NIX), belong to the BH3-only domain proteins of the BCL2 family, which are localized at MOM and are proapoptotic proteins [141].

Nix/BNIP3L anchors at MOM by its carboxy-terminus, whereas its amino terminus faces the cytoplasm, which contains an LC3-interacting region (LIR, SWxxL) capable of interacting with GABARAP and LC3 [142]. Nix/BNIP3L was reported to be primarily involved in stress sensing and in the induction of cell death when cellular stress prevailed [141]. Besides that, Nix/BNIP3L plays a key role during erythrocyte maturation by eliminating all the mitochondria by Ulk1-dependent but Atg5- and Atg7-independent mitophagy, where LIR–LC3 interactions guide the delivery of mitochondria to the autophagosome [143,144,145,146]. Additionally, Nix/BNIP3L has now been shown to induce mitophagy in several different type of cells, such as in neurons during ischemic stroke or cerebral ischemia [147,148]. Another possible mechanism proposed that oxidative phosphorylation (OXPHOS) stimulation induces Nix/BNIP3L-dependent mitophagy through mitochondrial translocation of a small GTPase RHEB (Ras homolog, Binds and activates mTORC1), which then promotes interactions between Nix/BNIP3L and LC3 to induce mitophagy [149]. Besides RHEB, the phosphorylation of Nix/BNIP3L (at serine 34 and 35) in close proximity to the LIR also promoted the Nix/BNIP3L–LC3 complex formation and enhanced the autophagosomal recruitment of mitochondria [150]. On the other hand, the phosphorylation of Nix/BNIP3L at Serine 212 by PKA inhibits mitophagy [151] (Figure 5A).

Like Nix/BNIP3L, another BH3-domain-containing protein, BNIP3, has been linked to mitophagy triggered by hypoxia in parallel to general autophagy. Hypoxia-induced autophagy was shown to be activated by both BNIP3 and Nix/BNIP3L through disrupting BCL-2-BECLIN1 interactions via binding to BCL-2 [152]. Additionally, BNIP3 could also activate autophagy by preventing the activation of the mTOR by sequestering the RHEB, the activator of mTOR that leaves mTOR inactivated [153]. BNIP3 shares 56% amino acid identity with Nix/BNIP3L [154] and both are upregulated by hypoxia-inducible factor 1 (HIF-1) that initiates LC3-dependent mitophagy during hypoxia [155,156]. Similar to Nix/BNIP3L, the LIR activity of BNIP3 requires phosphorylation of its serine 17 and 24, bordering the LIR domain by an unknown kinase and homodimerization of BNIP3 [157] (Figure 5B). Even after many similarities, the NIX/BNIP3L could not fully compensate for BNIP3 depletion to trigger mitophagy and cell death in response to ischemic stroke [147]. Besides inducing ubiquitin-independent mitophagy, both BNIP3 [158,159] and BNIP3L [160] can participate in ubiquitin-dependent mitophagy also by promoting the mitochondrial translocation of PARKIN and facilitating PARKIN-mediated mitophagy. In fact, Nix/BNIP3L has been shown to be a part of the PINK1–PARKIN pathway, as it was ubiquitinated by PARKIN, which recruited the mitophagy receptor NBR1 for mitophagy [161]. These studies suggested that mitophagy pathways could crosstalk with each other. However, evidence from cellular [162] and human studies [163] suggest that the PINK1–PARKIN pathway may not to be required for NIX/BNIP3L mediated mitophagy. This is evident from the study showing that NIX/BNIP3L induced mitophagy preserved the mitochondrial function in an asymptomatic homozygous *PARKIN* mutation carrier (lacking functional PARKIN) that protected the carrier from developing PD by her eighth decade [163]. Moreover, genetic and pharmacological induction of NIX/BNIP3L can restore mitophagy in skin fibroblasts from PD patients carrying mutations in *PINK1* or *PARKIN* [163]. More importantly, studies have demonstrated upregulation of Nix/BNIP3L and BNIP3 as a potential drug target in models of neurodegenerative diseases such as AD and PD. In that context, the NAD^+^ precursor restored the cognitive function in a *Caenorhabditis elegans-based AD* model by the activation of neuronal mitophagy mediated by BNIP3 equivalent DCT-1 [164]. Similarly, the NAD^+^ precursor rescued mitochondrial defects by upregulating Nix/BNIP3L expression in PD patient-isolated iPSC-derived neurons [165]. In agreement with neuroprotective role of Nix/BNIP3L, decreased Nix/BNIP3L mediated mitophagy was found to be detrimental for synaptic density and memory function in hippocampal neurons and in mice exposed to stress-related glucocorticoids, which are major etiological factors in the development of neurodegenerative diseases [166]. In contrast, NIX/BNIP3L upregulation by phorbol 12-myristate 13-acetate (PMA) pretreatment improved synaptic and cognitive function in glucocorticoids (corticosterone)-exposed mice [166]. These studies suggest the activation of alternative mitophagy pathways could be potential drug targets in neurodegenerative diseases.

#### 3.1.2. BCL2L13 in Mitophagy

BCL2-like 13 (BCL2L13) was discovered as a mitophagy receptor while screening for the mammalian ortholog of yeast Atg32; it is an essential mitophagy receptor in yeast [167,168]. Mitophagy protein Atg32 is a transmembrane protein embedded in the MOM that interacts with ubiquitin-like protein Atg8 through its Atg8-interacting motif (AIM) for the recruitment of autophagosomes to mitochondria [167]. In the process, Atg32 cooperates with scaffold protein, Atg11, that probably recruits the core Atg proteins to mitochondria undergoing mitophagy [167]. Similarly, BCL2L13 is also located on the MOM with its single transmembrane region and has the ability to bind to microtubule-associated protein 1A- or 1B-light chain 3B (LC3B), a mammalian ortholog of Atg8, through an LC3-interacting region (LIR) containing a WXXI motif, on its N-terminus facing the cytoplasm [168]. Moreover, exogenous BCL2L13 expression has the ability to partially restore mitophagy defects in Atg32-deficient yeast, indicating that BCL2L13 is indeed a functional mammalian ortholog of Atg32 [168]. Similar to yeast Atg32, in mammalian cells, the expression of BCL2L13 increased upon mitochondrial depolarization, which induced mitochondrial fragmentation and mitophagy in HEK293 cells [168]. In fact, BCL2L13 shares several molecular characteristics with its yeast counterpart Atg32, such as mitochondrial localization, WXXL or WXXI motifs, acidic amino acid clusters, and single membrane-spanning topology. Moreover, the phosphorylation of Ser272 on BCL2L13 stimulated the binding of BCL2L13 to LC3, similar to the case of Atg32 at Ser114 [168,169]. However, the mitophagy machinery, which Atg32 uses in yeast, may not be used by its mammalian counterpart BCL2L13, as evident from the differential requirement of Atg11 and Atg13. Atg11 was essential for Atg32-mediated mitophagy, whereas Atg11 was not required in BCL2L13-mediated mitophagy in yeast [167,170]. On the contrary, Atg13 is not essential for Atg32-mediated mitophagy, but it is indispensable in BCL2L13-mediated mitophagy in yeast [167,170]. Notably, BCL2L13 is able to promote mitochondrial fragmentation in DRP1-depleted cells, as well as mitophagy in *PARKIN*-deficient cells, indicating that the ubiquitination of mitochondrial proteins is not involved in BCL2L13-mediated mitophagy [168]. However, the molecular mechanism by which BCL2L13 coordinates mitochondrial fission and mitophagy was not described [168]. Later, the same group proposed that the ULK1 complex (composed of ULK1, ATG13, FIP200 and ATG101) which is involved in starvation-induced autophagy, is vital for BCL2L13-mediated mitophagy in mammalian cells [170]. Their study suggested that the BCL2L13 recruits the ULK1 complex to process mitophagy post or parallel to LC3B recruitment to mitochondria (Figure 5C). The interaction of LC3 with ULK1, as well as with BCL2L13, is important for mitophagy [170]. Recently, a specific role of BCL2L13 in the maintenance of mitochondrial quality via increased mitochondrial turnover synchronized with increased fusion and decreased fission was shown in humans as the long-term effects of chronic training or exercise [171]. 

#### 3.1.3. FKBP8 in Mitophagy

Like BCL2L13, FK506-binding protein 8 (FKBP8) was also identified as a mitophagy receptor during the screening for mammalian orthologs of yeast Atg32 [137]. The FKBP8 was able to promote stress-induced mitophagy in a PARKIN-independent manner [137]. 

FKBP8 is a MOM protein belonging to the FKBP family and, unlike other members of the FKBP family, it exhibits peptidylprolyl isomerase activity upon binding to calmodulin [172]. Besides that, FKBP8 also plays a role in anchoring the proteasome onto the mitochondria and it inhibits apoptosis by binding to BCL-2 [140,173]. The mitophagy receptor FKBP8 contains an LIR motif (like other mitophagy receptors) at its N-terminus and mediates mitophagy by interacting with LC3A that recruits autophagosome to mitochondria [137]. Overexpression of FKBP8 promotes mitochondrial fission and TOM20 degradation [137]. Moreover, the coexpression of FKBP8 with LC3A in HeLa cells induced PARKIN-independent mitophagy without depolarization [137]. Interestingly, in this setting, the majority of FKBP8 escaped the mitophagy-associated degradation, as shown earlier during CCCP-induced PARKIN-dependent mitophagy to prevent unwanted apoptosis [174]. Later investigations revealed that FKBP8 played an important role also in iron depletion- and hypoxia-induced mitophagy in mammalian cells [175] (Figure 5D). Under these stress conditions, FKBP8 played a dual role in mitochondrial fragmentation and mitophagy, with the help of its LIR motif-like sequence (LIRL) and the LIR motifs, respectively. FKBP8-induced mitochondrial fragmentation was independent of Drp1, BNIP3 and NIX but required OPA1, to which FKBP8 binds with its N-terminal LIRL motif, thus facilitating mitochondrial fragmentation and mitophagy [175]. However, the study found that FKBP8 knockdown did not affect mitochondrial fragmentation triggered by the treatment with CCCP, but it did affect mitochondrial fragmentation under hypoxic conditions [175], suggesting that mitochondrial fragmentation depends on different mediators in a stress-dependent manner. 

#### 3.1.4. BCL-XL as an Inhibitor of Mitophagy

The prosurvival members of the BCL-2 family (e.g., BCL-XL and MCL-1) have been found to suppress different mitophagy pathways, in sharp contrast to proapoptic BCL-2 family proteins (BNIP3 and Nix). BCL-2 proteins (e.g., BCL-XL and MCL-1) suppressed mitophagy through the inhibition of PARKIN translocation to depolarized mitochondria [176], thereby blocking PARKIN-dependent ubiquitination of mitochondrial substrates and downstream events. Interesting, this suppressive effect of BCL-XL and MCL-1 was found to be BECLIN-1-independent. BECLIN-1, the mammalian ortholog of yeast Atg6, has a functional BH3 domain. Therefore, BCL-2, BCL-XL, BCL-W and MCL-1 all bind to BECLIN-1 and prevent it from forming the initiation complex, which inhibits autophagy [177,178], partly explaining the mitophagy suppressive effect. The suppressive effect of BCL-2 proteins was mainly attributed to the continuous retrotranslocation of PARKIN from mitochondria to the cytosol, that blocked the feed-forward amplification loop of the PINK1–PARKIN pathway [176]. Similarly, another antiapoptotic protein BCL-B suppresses mitophagy in hepatic stellate cells by inhibiting the phosphorylation of PARKIN, as well as by directly binding to phospho-PARKIN. Suppression of mitophagy in this case also inhibited apoptosis [179]. 

Later studies also demonstrated that BCL-XL could inhibit PINK1/PARKIN-dependent mitophagy by preventing the accumulation of PARKIN on mitochondria via two ways: first by directly binding to PARKIN in the cytoplasm to prevent the translocation of PARKIN from the cytoplasm to mitochondria. Secondly, by binding to PINK1 on mitochondria to inhibit the signal for PARKIN translocation [180]. Contrary to PARKIN accumulation on mitochondria, the mitochondrial localization of YFP-PINK1 was not inhibited by BCL-XL under CCCP treatment [180]. 

BCL-XL not only affects the PINK1-PARKIN mitophagy pathway, but it can regulate FUNDC1 mitophagy also. It was found that BCL-XL interacted with PGAM5 to inhibit dephosphorylation of FUNDC1 and subsequent mitophagy [181]. Later, the same group found that the reciprocal interaction of PGAM5 with FUNDC1 and BCL-XL, was controlled by PGAM5 multimerization, which serves as a molecular switch between mitofission/mitophagy and apoptosis [182]. 

Taken together, it appears that pro-apoptotic signaling proteins (BNIP3, Nix) stimulate mitophagy whereas anti-apoptotic BCL-2 family members (e.g., BCL-XL, MCL-1) suppress mitophagy. However, such a hypothesis could be challenged, for example, by the FKBP8, which is an anti-apoptotic protein but stimulates mitophagy [137]. 

### 3.2. Other Mitophagy Receptors

#### 3.2.1. FUNDC1

One of the most prominent mitophagy receptors is FUN14 domain containing 1 (FUNDC1), which is located on the outer mitochondrial membrane with its three transmembrane domains. FUNDC1 is reported to regulate mitochondrial clearance in different mitochondrial stresses or physiological demands, such as in mitochondrial uncoupling or hypoxia-mediated mitophagy and paternal mitochondrial clearance in *C. elegans* [135,183,184]. Like other mitophagy receptors described above, FUNDC1 also executes mitophagy by directly interacting with LC3 via its conserved LC3-interacting region (LIR) facing the cytosol [135]. Similar to Nix/BNIP3L and BNIP3, FUNDC1 is also associated with hypoxia-induced mitophagy. Its deficiency blocks hypoxia-induced mitophagy that can then be restored only by wild-type FUNDC1, but not by LIR-mutated FUNDC1 [135]. Unlike NIX/BNIP3L and BNIP3, the expression of FUNDC1 does not vary substantially during hypoxic or mitochondrial depolarization events, possibly due to the lack of conserved HIF-1 sites in the promoter region of FUNDC1 [185]. Instead, the mitophagy activity of FUNDC1 is manipulated by several post-translational modifications. At the basal level, FUNDC1 is constitutively phosphorylated at tyrosine 18 and serine 13 by non-receptor tyrosine kinase (SRC) and casein kinase 2 (CK2), respectively, which reduces its interaction with LC3 [135,183]. Upon hypoxia or mitochondrial depolarization, FUNDC1 dephosphorylations at tyrosine 18 and serine 13 are carried out by the mitochondrial phosphatase PGAM family member 5 (PGAM5). Concomitantly, the FUNDC1 is phosphorylated at serine 17 by ULK1. These post-translational modifications promote the interaction of FUNDC1 with LC3, which results in mitophagy [31,135,183] (Figure 6A). 

Other than phosphorylation and dephosphorylation, mitochondrial E3 ubiquitin ligase MARCH5-mediated ubiquitination of FUNDC1 at K119 has been found to inhibit initial hypoxia-induced mitophagy via the proteasomal degradation of FUNDC1 [186]. The mitophagy activity of FUNDC1 is additionally fine-tuned by BCL2-like 1 (BCL2L1 better known as BCL-XL) via regulating the activity of PGAM5 [181]. During homeostasis, PGAM5 activity is inhibited by BCL-XL, but under hypoxia BCL-XL gets degraded, leading to the activation of PGAM5 that promotes the dephosphorylation of FUNDC1, resulting in the induction of mitophagy [181]. Later the same group proposed that dimeric or multimeric states of PGAM5 regulate its interaction with BCL-XL and FUNDC1 reciprocally, which may serve as a molecular switch between mitophagy and apoptosis [182]. At the basal level, PGAM5 interacts with BCL-XL and dephosphorylates it to inhibit apoptosis. However, under oxidative stress conditions, PGAM5 assumes a multimeric state and releases BCL-XL, leading to an increase in BCL-XL’s phosphorylation and ultimately to apoptosis. Once liberated from BCL-XL sequestration, multimeric PGAM5 is able to dephosphorylate FUNDC1, to augment mitochondrial fission and induce mitophagy [182] (Figure 6A). As a future direction of research, it would be interesting to investigate the role of Keap1 in multimerization of PGAM5 under oxidative stress since Keap1 is an important ROS sensor as well as PGAM5 interacting protein [187,188]. Recent studies have identified additional factors affecting FUNDC1 via PGAM5, such as syntaxin 17 (STX17), a SNARE protein located in the mitochondria-associated membranes (MAM) and mitochondria, is found essential for dephosphorylation of FUNDC1 by PGAM5 during mitophagy [189]. 

Interestingly, like Nix/BNIP3L and BNIP3, FUNDC1 could also be involved in PINK1–PARKIN dependent mitophagy, since depletion of FUNDC1 not only inhibited CCCP-induced mitochondrial clearance but also prevented the whole coverage of mitochondria by PARKIN and mitochondrial aggregation observed after CCCP treatment [189]. These findings again suggest the possibility of cross-talk between different mitophagy pathways. In that context, it would be interesting to investigate if BNIP3 or Nix/BNIP3L mitophagy can compensate for the depletion of FUNDC1 and vice versa under common mitochondrial stresses, such as hypoxia or CCCP treatment.

Moreover, studies have shown that FUNDC1 controls mitochondrial dynamics in relation to FUNDC1-mediated mitophagy, by controlling the integrity and function of mitochondria-associated ER-membranes (MAMs), the functional and physical contacts between mitochondria and ER [185,190]. According to the study, the FUNDC1 accumulates in MAMs with the help of the ER protein calnexin (CANX), but during hypoxia the association between FUNDC1 and CANX perishes and the interaction between FUNDC1 and DRP1 prevails, which brings DRP1 at MAMs to trigger mitochondrial fission and mitophagy [190]. In another study, FUNDC1 was found to interact with the mitochondrial fission factor DRP1 and inner membrane fusion regulator OPA1 both to coordinate mitochondrial dynamics and mitophagy [191]. Mitophagy stresses, such as selenite, FCCP treatment or FUNDC1 dephosphorylation, cause the disassembly of the FUNDC1–OPA1 complex, while enhancing the association of FUNDC1 with DRP1, leading to mitochondrial fission, thus fostering mitophagy [191]. 

Pathophysiologically, FUNDC1 has been shown to play an important role in metabolic & cardiovascular diseases, obesity and cancer. In cardiovascular and metabolic diseases, FUNDC1 is generally considered to play a protective role, since FUNDC1-mediated mitophagy can alleviate mitochondrial damage caused by intracellular stresses such as hypoxia and ischemia/reperfusion (I/R), thus benefiting cellular health overall [192,193]. The protective role of FUNDC1 has also been reported in pathological conditions, including heart failure, cardiac aging, myocardial infarction, diabetes or obesity-associated complications [185,194,195]. In contrast, there are reports suggesting the detrimental role of FUNDC1 on cardiac function, such as in diabetic patients. Diabetes was shown to suppress AMP-activated protein kinase that initiated FUNDC1-mediated MAM formation, resulting in mitochondrial dysfunction and cardiomyopathy [196]. In the brain, FUNDC1 alleviated intracerebral hemorrhage-induced brain injury by promoting mitophagy, which blocks inflammation via inhibiting the NLRP3 inflammasome pathway in mice [197].

In contrast to metabolic & cardiovascular diseases and obesity, the roles of FUNDC1, vary significantly among different cancers. Although, a positive correlation between FUNDC1 expression and tumor progression can be assumed as hypoxia, a microenvironmental characteristic of cancers, might induce FUNDC1-dependent mitophagy. Mitophagy has been reported to promote tumor cell survival earlier by preventing apoptosis or improving overall mitochondrial function [198,199,200,201]. Additionally, FUNDC1 has been found to assist angiogenesis by increasing Ca^2+^ in the cytosol by MAMs formation that increased vascular endothelial growth factor receptor production. In contrast, FUNDC1 deficiency inhibited angiogenesis that reduced tumor size significantly in mice [202]. Thus, FUNDC1 was shown to enhance the progression of cancer and represented a poor prognosis in some tumors [203,204]. Accordingly, FUNDC1 exhibited detrimental effects on skin cancers, head and neck cancers, cervical cancer [204], breast cancers [205] and liver hepatocellular carcinoma [206]. 

In contrast to the above, there are several reports suggesting FUNDC1 could inhibit carcinogenesis by inducing mitophagy, and could display protective roles in hepatocellular carcinoma [207], laryngeal cancers [208], ovarian cancers, bladder cancers and lung cancers [206]. The FUNDC1 has been implicated in mitochondrial reprogramming and cellular plasticity in cancer, with the help of mitochondrial matrix protease LonP that may potentially antagonize tumor growth [203]. Another study on a mouse model of human hepatocellular carcinoma (HCC), induced by the chemical carcinogen diethylnitrosamine (DEN), suggested that FUNDC1 suppresses HCC initiation by increasing mitophagy that reduced inflammasome activation and inflammatory responses in hepatocytes [207]. 

Overall, the role of FUNDC1-mediated mitophagy may differ in different stages of tumor development. In the early stage of tumorigenesis, mitophagy maintains normal cell metabolism and inhibits tumorigenesis, while in the later stage of tumor development, the occurrence of FUNDC1 mitophagy improves cell tolerance and promotes the development of the tumor [207,209].

#### 3.2.2. Prohibitin 2 (PHB2) in Mitophagy

Prior studies on mitophagy receptors were focused on the mitochondrial outer membrane proteins until recent identification of PHB2 (Prohibitin 2), the inner mitochondrial membrane protein, as a novel mitophagy receptor by Wei et al. [210] (Figure 6B). Unlike other mitophagy receptors, PHB2 in fact requires PARKIN for mitophagy induction [210]. PHB2 is a component of the mitochondrial prohibitin complex, along with PHB/PHB1 (prohibitin). PHB2 and PHB assemble into a ring-like macromolecular structure consisting of 12–16 heterogenous dimer pairs to form the prohibitin complex at the mitochondrial inner membrane. The prohibitin complex binds and regulates the activity of mitochondrial m-AAA protease, which controls mitochondrial membrane protein processing and degradation [211]. 

Wei et al. demonstrated that PHB2 assisted mitochondrial clearance by interacting with LC3-II upon treatment with CCCP or oligomycin and antimycin (OA) in HeLa cells that stably express PARKIN, but not in HeLa cells lacking *PARKIN* [210]. PHB2 directly interacted with LC3-II through its LIR motif, whereas PHB associates with LC3-II indirectly via PHB2. PHB2 could access LC3-II only after disruption of the MOM achieved by PARKIN-mediated ubiquitination of MOM proteins and their subsequent degradation by the proteasome [71]. Knockdown or conditional knockout of *PHB2* in HeLa cells inhibited mitochondrial clearance in OA-treated cells, but did not prevent MOM rupture, indicating that MOM degradation preceded PHB2-LC3-II interaction and mitochondrial clearance [210]. Moreover, mutation in the LIR motif of *PHB2* precluded LC3 binding without compromising its other functions, but resulted in a reduced mitochondrial clearance upon OA treatment in mouse embryonic fibroblasts [210]. Inhibition of mitophagy by PHB2 knockdown was similar to the phenotype demonstrated by ATG7 knockdown and suggested that mitophagy is specifically affected [210]. The physiological importance of the PHB2 was demonstrated in *C. elegans* where knockdown of PHB2 in males led to the retention of paternal mitochondrial DNA in later stages of embryonic development and subsequent generation [210]. 

Later studies reported that PHB2 forms a ternary protein complex with sequestosome 1 (p62/SQSTM1) and LC3 in bile acid-mediated mitophagy in L02 liver cells and was required for cholestasis-induced mitophagy in the liver [212]. Subsequently, PHB2 was reported to be involved in platelet mitophagy to maintain the stability of the mitochondrial structure in platelets. The PHB2 mediated mitophagy plays an important role in platelet activation by downregulating the expression of platelet activation genes in MEG-01 cell line [213]. 

A recent study demonstrated a new signaling pathway for PHB2 in which it can regulate PINK1–PARKIN-mediated mitophagy by controlling the stabilization of PINK1 through the PARL–PGAM5 axis [214] (Figure 6B). The study reported that PHB2 depletion blocked the stabilization of PINK1 and the recruitments of PARKIN, ubiquitin and OPTN, to mitochondria, leading to an inhibition of mitophagy upon mitochondrial membrane depolarization or misfolded protein aggregation [214]. However, overexpression of PHB2 or PHB2 mLIR (unable to bind to LC3) restored PINK1 stabilization and PARKIN recruitment to mitochondria in PHB2-deficient cells, indicating that the PHB2-regulated PINK1–PARKIN mitophagy is independent of its LC3 binding ability. Additionally, overexpression of PHB2 or PHB2 mLIR induced PARKIN recruitment to the mitochondria, as well as PARKIN-dependent mitophagy [214]. The observed effects were suggested to be mediated by PHB2 negatively regulating the activity and stability of the mitochondrial protease PARL, which regulates the processing of PINK1 and its interactor PGAM5. Thus, PHB2 depletion accelerated PARL activity, which led to increased processing of PGAM5 and PINK1, which resulted in non-stabilization of PINK1 and inhibition of PARKIN translocation [214]. Interestingly, the study reported that PHB2 could interact with both PARL and PGAM5, but these interactions correlated inversely upon CCCP treatment. The CCCP treatment increased PHB2–PARL interaction, whereas it decreased the PHB2–PGAM5 interaction. Increased PHB2–PARL interaction inhibited PARL activity that decreased the processing of both PINK1 and PGAM5, which resulted in PINK1 stabilized on MOM. The study suggested that only the full length PGAM5, not its PARL-cleaved form, helped PINK1 stabilize on MOM [214]. Therefore, inhibition of PGAM5 processing by PARL additionally supported PINK1 stabilization besides PHB2-mediated PARL inhibition. According to the study PHB2 controls the rupture of the mitochondrial outer membrane also by regulating PINK1–PARKIN mitophagy. Once the outer mitochondrial membrane is ruptured, PHB2 would further facilitate mitophagy by directly binding to LC3 (Figure 6B) [214]. A ligand of PHB proteins was synthesized, called FL3, which strongly inhibited PHB2-mediated mitophagy and effectively blocked cancer cell growth at nanomolar concentrations. Thus, their findings revealed that the PHB2–PARL–PGAM5–PINK1 axis is a novel pathway of PHB2-mediated mitophagy [214]. However, so far it is not known if the mitophagy receptors from the inner and outer mitochondrial membrane function synergistically to promote mitophagy.

#### 3.2.3. AMBRA1 as a Mitophagy Receptor

The Activating Molecule in BECLIN1-Regulated Autophagy1 (AMBRA1) has been reported to enhance mitochondrial clearance after mitochondrial depolarization (by CCCP) via interacting with PARKIN [215]. Though PARKIN interacts with AMBRA1 it did not ubiquitinate AMBRA1, which redistributes around depolarized mitochondria in a PARKIN dependent manner and activates class III phosphatidylinositol 3-kinase (PI3K) complex essential for autophagy [215]. However, the localization of AMBRA1 to mitochondria is not enough to activate mitophagy, as previous studies demonstrated a pool of AMBRA1 localized to the mitochondria, but its pro-autophagic activity was kept inhibited by the mitochondrial resident BCL-2 [216]. Following the induction of autophagy AMBRA1 is released from the mitochondrial BCL2 to interact with BECLIN1 and participate in the autophagy initiation [216]. AMBRA1 emerged as a major mitophagy player after a study describing it as an additional mitophagy receptor which could directly mediate mitophagy by interacting with LC3 via its C-terminally located LIR motif that could amplify PARKIN-mediated mitochondrial clearance, as well as regulate PARKIN-independent mitophagy [217]. Moreover, forced mitochondrial AMBRA1 (AMBRA1^ActA^) expression resulted in mitochondrial network redistributed as mito-aggresome-like structures around the perinuclear area, which colocalized with ubiquitin and subsequently with LC3, resulting in massive mitophagy [217]. Moreover, high levels of AMBRA1 at the mitochondria were sufficient to induce PARKIN- or p62-independent mitophagy that required only LC3 binding via the LIR motif of AMBRA1 [217]. Not only mitochondrial AMBRA1 but non-targeted AMBRA1 expression could also induce mitophagy through its LIR domain in the absence of the PINK1–PARKIN post-FCCP treatment [217]. Moreover, AMBRA1 expression can partially rescue mitochondrial clearance in *PINK1*-deficient PD patient’s fibroblasts [217]. Later, AMBRA1 was found sufficient to restore mitophagy in fibroblasts of PD patients carrying mutations in *PINK1* or *PARKIN* [218]. The study demonstrated that mitochondrial AMBRA1^ActA^ over-expression induced mitophagy in SH-SY5Y that offered neuroprotection by suppressing oxidative stress and apoptosis induced by Parkinsonian neurotoxins 6-OHDA and rotenone [218]. 

Subsequently, a detailed study revealed the mechanism of AMBRA1 mediated mitophagy by demonstrating that mitochondrial-targeted (AMBRA1^ActA^) or non-targeted (AMBRA1^WT^) expression of AMBRA1 could promote the translocation of E3 ubiquitin ligase HUWE1 to the mitochondria, which enhanced the ubiquitination of mitochondrial proteins and phosphorylation of AMBRA1 near its LIR motif (S1014), resulting in mitophagy induction [219]. Similar to other mitophagy receptors, phosphorylation of AMBRA1 near its LIR motif regulated the AMBRA1–LC3B interaction and its mitophagy activity. AMBRA1 phosphorylation is regulated by an upstream kinase, IκBα kinase (IKKα), the key player in inflammation [220]. The study revealed that AMBRA1 could act as a physiologically relevant mitophagy receptor, as it was found capable to induce mitophagy in Penta-KO cells (NDP52-, OPTN-, TAX1BP1-, NBR1- and P62-deficient) and to regulate ischemia-induced mitophagy, together with HUWE1 and IKKα [219]. However, a question remained unanswered in the study; how does E3 ubiquitin ligase HUWE1 influence the phosphorylation of AMBRA1 by IKKα? 

## 4. LIPID-Based Mitophagy

Certain classes of lipids that have the ability to relocate to MOM under mitochondrial stress can also directly interact with LC3. This lipid–protein interaction at the MOM facilitates the recruitment of autophagosomes to initiate mitophagy. The following description deals with known lipids as mitophagy receptors.

### 4.1. Cardiolipin as a Mitophagy Receptor

Specific mitochondrial lipids can also function as mitophagy receptors, such as cardiolipin (CL), which is a mitochondrial inner membrane (MIM) phospholipid. Cardiolipin can relocate to the mitochondrial outer membrane to serve as a mitophagy receptor by directly interacting with LC3 upon mitochondrial stress or damage [98,221,222]. CL makes up to 20% of the total mitochondrial phospholipid content but it is distributed asymmetrically between the MIM (~97%) and MOM (~3%) [223,224,225]. CL adopts a cone-shaped geometry, owing to its flexible four fatty acyl chains bound to a glycerol head group, making it a suitable phospholipid for high membrane curvature regions such as the mitochondrial inner membrane [223]. CL is required for the optimal function of the mitochondrial electron transport chain, ADP/ATP translocase involved in ATP synthesis, ROS production and apoptosis [226,227,228]. Besides these functions CL is also reported as mitophagy receptor since the mitochondrial injuries by mitochondrial toxins rotenone, staurosporine or 6-hydroxydopamine induced the translocation of the CL from MIM to the MOM in neuroblastoma cells and primary cortical neurons [98,221] or in CCCP-treated murine lung epithelial MLE-12 cells and human cervical adenocarcinoma HeLa cells [222]. At the MOM, the CL directly recruits LC3 to mitochondria by binding to the N-terminal helix of LC3 independently of mitochondrial depolarization to induce mitophagy [98,222] (Figure 7A). As an alternative outcome, the externalized CL oxidation could result in activation of apoptotic steps, such as mPTP opening and cytochrome c release from mitochondria [229,230,231]. So far, two molecules are reported to be essential for the externalization of CL to MOM; first is the mitochondrial phospholipid scramblase-3 (PLSCR3), and second is the hexameric intermembrane space protein complex of mitochondrial nucleoside diphosphate kinase D (NDPK-D) [98,222]. The CL-transfer activity of NDPK-D is closely associated with the dynamin-like GTPase OPA1, implicating fission–fusion dynamics in CL-mediated mitophagy, similar to all other mitophagy mechanisms [222]. In fact, mitochondrial fission protein DRP1 binds to CL via its B-insert domain that stimulates DRP1 oligomerization and its GTPase activity, enhancing mitochondrial fission [232,233,234,235]. This designates CL at the MOM as a pro-fission phospholipid, whereas at MIM, CL stimulates OPA1-mediated inner membrane fusion by heterotypic interaction [236,237]. In contrast, conversion of CL to phosphatidic acid (PA) at MOM by the mitochondrial phospholipase D [238] negatively regulates DRP1-dependent mitochondrial fission [239] and could enhance Mfn1/2-dependent MOM fusion [240]. Mitochondrial fission could facilitate mitophagy, since large dysfunctional mitochondria (5 μm) must be engulfed by relatively smaller autophagosomes (1 μm) for degradation [221].

A recent report demonstrated that externalized CL binds preferably to LC3A, as it has a higher affinity when compared to that of LC3B, but both proteins showed a similar ability to colocalize with mitochondria upon induction of CL externalization by rotenone in HeLa-NDPK-D or SH-SY5Y cells. Moreover, the in vitro results suggested a possible role of LC3A, but not of LC3B, in oxidized-CL recognition as a counterweight to excessive apoptosis activation [241].

Externalization of CL to MOM was also promoted in dopaminergic neurons (hiPSC derived) by A53T and E46K mutations in *SNCA* gene (α-synuclein, α-syn) linked to autosomal dominant familial PD. The externalized CL pulled α-syn monomers away from toxic oligomeric fibrils [242,243] and facilitated their refolding from aggregated β-sheet forms back to monomers comprising α-helices [244] to slow down α-syn pathogenicity [245]. In parallel, the binding of A53T and E46K α-syn to CL triggered excessive mitophagy by increasing the recruitment of LC3 to the MOM [245]. Thus, externalized CL at MOM appears to handle two PD-associated neurodegenerative mechanisms, the protein aggregation and mitochondrial dysfunction, simultaneously.

The mitophagy signaling by externalized CL has not only been demonstrated in in vitro studies but it has been found to act as an early mitophagy trigger after traumatic brain injury (TBI) in human and rat brain samples [34]. Mitophagy after TBI is suggested to be an attempt to avoid further neuronal damage from irreversible neural apoptosis and ROS overproduction [34]. The externalization of CL was found to be dependent on PLSCR3 and CL synthase. The increased phosphorylation of PLSCR3 correlated with the higher levels of CL in MOM fractions in injured brain samples [34]. Phosphorylation of Thr21 in PLSCR3 by protein Kinase C-delta [246] has been reported to enhance the transport of CL from MIM to MOM. 

The externalized CL could further assist mitophagy through its direct interaction with the autophagy protein BECLIN1, since the evolutionarily conserved domain of BECLIN1 preferentially binds lipid membranes enriched in CL [221]. 

### 4.2. Ceramide as a Mitophagy Receptor

Other evidence confirming that, indeed, lipids could act as mitophagy receptors came from studies demonstrating that mitochondrial ceramides or their analogues induce lethal mitophagy and tumor suppression in head and neck squamous cell carcinoma cells [247,248] and acute myeloid leukemia cells [249] in vitro and in vivo. 

The sphingolipid, ceramides are composed of a sphingosine backbone and a fatty acyl chain. Their de novo synthesis depends on fatty-acyl CoA and six different ceramide synthases (named from CerS1 to CerS6), which are integral endoplasmic reticulum (ER) membrane proteins. Each of them synthesizes ceramides with acyl chain lengths ranging from 14 to 26 carbons, having distinct biological functions [250,251]. For example, CerS1 and CerS6 preferentially generate C_18_- and C_16_-ceramide, respectively. Mass spectrometry identified 31 mitochondrial sphingolipids, including ceramides, glyceroceramides (GlcCer), sphingomyelins (SM), and gangliosides (Gan) [252]. Notably, ceramides have emerged as key mediators of anti-proliferative and tumor-suppressive cellular programs, such as apoptosis, mitophagy, cell cycle arrest and senescence [251]. Multiple stimuli, including tumor necrosis factor α (TNFα) [253], ionizing radiation [254] and chemotherapeutic drugs [255], cause a rise in ceramide levels through stimulation of de novo ceramide synthesis or by sphingomyelin hydrolysis, or both. Ceramides’ accumulation at MOM form channels there, affecting mitochondria function and homeostasis.

C_18_-ceramide generation by CERS1 expression in multiple head and neck squamous cell carcinoma (HNSCC) cell lines, or their treatment with the mitochondria-accumulating ceramide analog D-e-_14_C_18_-pyridinium ceramide bromide, selectively induced lethal mitophagy independently of apoptosis [247,248]. Contrary to CerS1, CerS6 overexpression in the HNSCC cell line neither caused C_16_-ceramide accumulation in mitochondria nor induced mitophagy. However, treatment of the cell line with the mitochondria-accumulating C_16_-ceramide analog C_16_-pyridinium ceramide-induced mitophagy, suggesting that the localization of ceramide species, rather than their length of the fatty acyl chain, is essential for ceramide-induced mitophagy [247]. To induce lethal mitophagy, the MOM localized ceramide or its analogue recruited LC3B-II to mitochondria by binding to it. This lipid–protein complex formation required Ile35 and Phe52 residues of the hydrophobic domain of LC3B, as well as PE lipidation [247]. Mitochondrial fission again played a key role in ceramide-induced mitophagy, as DRP1 knockdown affected ceramide distribution in MOM that prevented mitophagy [247]. 

Recently, the mechanism underlying ceramide-mediated mitophagy was demonstrated in human HNSCC UM-SCC-22A cells [256]. According to the proposed mechanism, stress signaling activates DRP1, which induces mitochondrial fission, accompanied by mitochondrial membrane damage. This is followed by the trafficking of newly translated CerS1 from the ER’s surface to MOM through mitochondria-associated membranes (MAMs) by the Protein that mediates ER–mitochondria Trafficking p17/PERMIT, by interacting with TOM20 [256]. Consequently, the mitochondrially translocated CerS1 starts C_18_-ceramide generation, which induces the LC3B-II-mediated targeting of autophagosomes to mitochondria for lethal mitophagy [247,256] (Figure 7B). 

Ceramide-mediated lethal mitophagy is suggested to be involved in tumor suppression in severe combined immunodeficient (SCID) mice carrying human papillomavirus (HPV)-positive UM-SCC-47-derived HNSCC xenograft tumors, in response to cisplatin [257]. The study suggested that HPV early protein 7 (E7) enhances ceramide-mediated lethal mitophagy in response to chemotherapy-induced cellular stress in HPV-positive HNSCC cells, by selectively targeting the retinoblastoma protein (RB). Inhibition of RB by HPV-E7 relieves E2F5, which then associates with DRP1, providing a scaffolding platform for DRP1 activation and mitochondrial translocation, leading to mitochondrial fission and increased lethal mitophagy. This finding is relevant to the observed improved survival in response to chemo-radiotherapy in patients with oropharynx head and neck squamous cell carcinoma (HNSCC) infected with human papillomavirus (HPV) [257].

The tumor-suppressive role of ceramide-induced mitophagy is also evident from the study in acute myeloid leukemia (AML), where FLT3-ITD (mutant Fms-like tyrosine kinase 3–internal tandem duplication) signaling downregulated the CerS1/C_18_-ceramide which confers its resistance to cell death [249]. Molecular or pharmacological targeting of FLT3-ITD reactivates CerS1/C_18_-ceramide generation, accompanied by mitochondrial division, mitophagy and cell death both in AML cell lines and blasts obtained from FLT3-ITD1 patients, and in xenograft models in vivo [249]. More direct evidence came from the results with a soluble C_18_-ceramide analog drug, LCL-461 which accumulated in mitochondria and induced lethal mitophagy in AML cells expressing FLT3 mutations and in blasts obtained from FLT3-ITD1 AML patients [249]. Again, DRP1 activation acted as a signal to translocate CerS1 to mitochondria [249].

Other than lethal mitophagy, ceramide induces protective mitophagy also, for example, the overexpression of CerS2 in human cardiomyocyte-like cell line (AC16) with elevated, very long chain ceramides (VLC), that caused insulin resistance, oxidative stress, mitochondrial dysfunction and mitophagy [258]. Inhibition of mitophagy exacerbated cell death, suggesting that CerS2 and VLC induced adaptive and protective mitophagy [258]. 

## 5. Micromitophagy

Depending on how mitochondria are delivered to the lysosome, mitochondrial elimination occurs via two pathways: macroautophagy and microautophagy (termed macromitophagy and micromitophagy, respectively) [23]. Macromitophagy is a selective form of macroautophagy, characterized by the formation of a double membrane vesicle termed an autophagosome, that ultimately transfers mitochondria to endosomes/lysosomes via fusion for degradation. Micromitophagy, in contrast, represents microautophagy, in which mitochondria are directly sequestered into endosomes/lysosomes for degradation in the absence of autophagosome formation [23,28,259]. Micromitophagy is the targeted removal of only the damaged part of mitochondria, instead of the entire organelle, by formation of mitochondria-derived vesicles (MDV) that bud off and then transit to lysosomes (Figure 7C). Such micromitophagy could play a crucial role in mitochondrial quality control in non-dividing post-mitotic cells such as neurons and cardiomyocytes, which cannot afford the complete mitochondrial removal as a means of mitochondrial quality control [260]. 

MDV generation and degradation now belong to mitochondrial quality control pathways that complement other MQC pathways when they are either overwhelmed or compromised [28,259]. In contrast to other mitophagy, MDV generation or micromitophagy operates independently of mitochondrial depolarization, autophagy signaling (Atg5, LC3, BECLIN-1), or DRP1-dependent mitochondrial fission [28]. However, PARKIN and PINK1 have been reported to be required for the biogenesis of MDVs that are devoid of TOM20 but contain other mitochondrial cargos that are trafficked to the lysosome [261]. Oxidative stress stimulates MDV formation, and the MDVs themselves are enriched in oxidized mitochondrial proteins [259]. A proposed mechanism for MDV generation involves the accumulation of protein aggregates in close proximity to mitochondrial membranes under oxidative stress conditions [262]. This event, concomitant with cardiolipin oxidation, would generate unconventional changes in the mitochondrial membrane structure such as curvatures [262]. The formation of mitochondrial membrane curvatures is thought to be followed by the accumulation of PINK1 at the MOM, followed by the ubiquitination and recruitment of PARKIN [262]. The process would eventually culminate in the formation of a vesicle, which is then released through a process involving unidentified proteins [262]. Electron microscopy evidence suggests that an MDV enters the lumen of multivesicular bodies, a form of late endosome, by invagination of its membrane. Subsequently, multivesicular bodies fuse with lysosomes to complete the hydrolytic degradation of the MDV [28]. 

Mitochondrial proteins are present in extracellular vesicles (EVs), which are secreted by most cells as communication signals even under unstimulated conditions [263]. The release of damaged mitochondrial components or mitochondrial DNA can act as damage-associated molecular patterns (DAMPs) that activate the innate immune system [264]. Therefore, to avoid the delivery of damaged mitochondrial components to EV, cells selectively regulate the packaging of mitochondrial protein into two distinct types of MDV. First, delivery of mitochondrial proteins to EVs requires Snx9-dependent MDVs, a subset of MDVs that were previously shown to regulate mitochondrial antigen presentation [265]. Second, MDVs carrying damaged mitochondrial components or oxidized cargo-enriched MDVs are targeted for lysosomal degradation in a process that depends on the PD-linked proteins PINK1 and PARKIN [265,266]. Through this alternative degradative route, mildly damaged mitochondrial components are processed and disposed within EVs of mitochondrial origin (mitochondrial-derived vesicles, MDVs) [262]. This mechanism contributes to organelle homeostasis before whole mitochondrial degradation is triggered [262].

Multiple levels of mitochondrial quality control exist to ensure mitochondrial plasticity, disposal and replenishment in order to maintain a well-functioning organelle network to meet the dynamic energy demands of the cell. It is possible that additional, currently unidentified pathways of mitochondrial quality control exist in cells. For example, a distinct endosomal–lysosomal mitochondrial degradation pathway exists that delivers PARKIN-wrapped entire mitochondria into RAB5A-positive early endosomes for lysosomal degradation through ESCRT (endosomal sorting complexes required for transport) machinery [267]. This mitophagy differs from MDV in several aspects such as mitochondrial size, dependence on BECLIN1 and early or late endosome delivery [267]. 

## 6. Mitophagy and Mitochondrial Biogenesis Could Be Linked 

Accumulating studies indicate that mitophagy and mitochondrial biogenesis could be closely interlinked. One of the first pieces of evidence comes from Parkinson’s disease-linked proteins PINK1 and PARKIN playing key roles in mitophagy degradation of mitochondria (already discussed above). Interestingly, they also regulate mitochondrial biogenesis by promoting the degradation of PARIS/ZNF746, a transcriptional repressor of the master regulator of mitochondrial biogenesis, PGC-1-α [109,110]. PINK1 interacts and phosphorylates PARIS at S322 and S613 to prime its ubiquitination by PARKIN and subsequent degradation [109,110]. Moreover, PINK1 and PARKIN control the localized translation of nuclear-encoded mRNAs, specific for respiratory chain components on the MOM [268]. A point to note is that PARKIN without phosphorylation is autoinhibited [269] until PINK1 is stabilized at mitochondria, which then phosphorylates PARKIN to activate it [270]. These events occur only if the mitophagy is in progress. This scenario suggests that PINK1–PARKIN mitophagy and mitochondrial biogenesis might proceed hand-in-hand to maintain an adequate functional mitochondrial pool. 

The second support for the hypothesis comes from the observation that during mitophagy, the Transcription factor EB (TFEB) translocates to the nucleus in a PARKIN- and PINK1-dependent manner [271]. The TFEB has been found to promote the expression of the master regulator of mitochondrial biogenesis, PGC-1α, to restore mitochondrial function and cell viability in human iPSC derived neurons [272], as well as in mouse liver during starvation that stimulate lipid metabolism [273]. 

Another PINK1–PARKIN mitophagy pathway player, PGAM5, has been found to stimulate mitochondrial biogenesis upon its release into the cytosol from mitochondria [274]. PGAM5 is a mitochondrial phosphatase and, similar to PINK1, it is cleaved by the PARL [44]. PINK1 and PGAM5 are suggested to be cleaved by PARL reciprocally, meaning that if PINK1 is stabilized then PGAM5 is preferably cleaved by PARL, and vice versa [275]. Mitophagy activation stabilizes PINK1, which might activate PGAM5 cleavage and its release into the cytosol [274,276]. Cleaved PGAM5 release from mitochondria could depend on the proteasome-mediated rupture of the outer membrane during mitophagy [276], which has been previously shown to precede the autophagic removal of remaining mitochondria [71]. It has been found that PGAM5 in cytosol interacts with the Wnt pathway component axin, and blocks axin-mediated β-catenin degradation by its dephosphorylation. This results in increase in β-catenin levels and β-catenin-dependent transcription [274]. The activation of Wnt/β-catenin signaling was found to stimulate the biogenesis of mitochondria, resulting in an increased mitochondrial number in C2C12 myocytes [274].

Other molecules have also been found to possibly link the two mechanisms, such as Adenosine Mono Phosphate Kinase (AMPK), which is shown to directly promote PGC-1α expression and mitochondrial biogenesis by phosphorylating epigenetic factors [277]. On the other hand, AMPK activation has also been found to promote mitochondrial fission and mitophagy in the skeletal muscle cell line C2C12 [278]. Moreover, AMPK-mediated phosphorylation of ULK1 is required to target mitochondria to the lysosome in exercise-induced mitophagy [279]. Taken together, AMPK appears to be involved in multiple aspects of mitochondrial physiology, including mitochondrial dynamics, mitophagy and mitochondrial biogenesis. 

## 7. Future Direction

The existence of a plethora of mitophagy mechanisms suggests the importance of this pathway in orchestrating mitochondrial quality control in cell physiology. However, it is currently unclear why such a large number of mitophagy pathways are required to regulate the same process. One explanation could be that such diversity allows the cell to respond appropriately to different situations. For example, FUNDC1 is activated to eliminate mitochondria in response to hypoxia, PINK1 and PARKIN for depolarization and NRF2-p62/SQSTM–KEAP1–RBX1 mitophagy pathway for oxidative stress. At the same time, it raises the question if only a particular mitophagy pathway will be activated in certain conditions, such as hypoxia, or whether multiple pathways could be activated simultaneously. Yet another possibility is that an alternative mitophagy pathway will be activated upon failure of a preferred pathway for the given condition. However, the possibility of such compensation is doubtful, as evident from the linkage of autosomal recessive PD with *PINK1* and *PARKIN* mutations. In this case, other mitophagy pathways do not compensate for the failure of the PINK1–PARKIN pathway. An alternative explanation could be the existence of specificity of the mitophagy pathway in a particular cell type. The addition of cell types or tissues adds up to another layer of complexity of mitophagy pathways, demanding further investigations. These investigations could significantly improve human health by providing drug targets, as defects in mitophagy contribute to various human disorders such as neurodegenerative diseases, cardiovascular diseases and liver diseases. Mitophagy research has increased the efforts to identify molecules capable of inducing mitophagy, which are currently at different levels of investigation, ranging from preclinical to clinical trials, such as in the case of Urolithin A [280]. Although the role of mitophagy in several neurodegenerative diseases is gradually becoming established, its role in cancer is far from established, demanding conclusive future investigation. Another fascinating aspect is the involvement of the same proteins such as DRP1 and PGAM5 in different mitophagy pathways, suggesting considerable crosstalk between different mitophagy pathways. Lipid-induced mitophagy is a relatively unexplored topic, specifically in regard to its importance in normal cell physiology and pathophysiology. 

## Figures and Tables

**Figure 1 cells-11-00038-f001:**
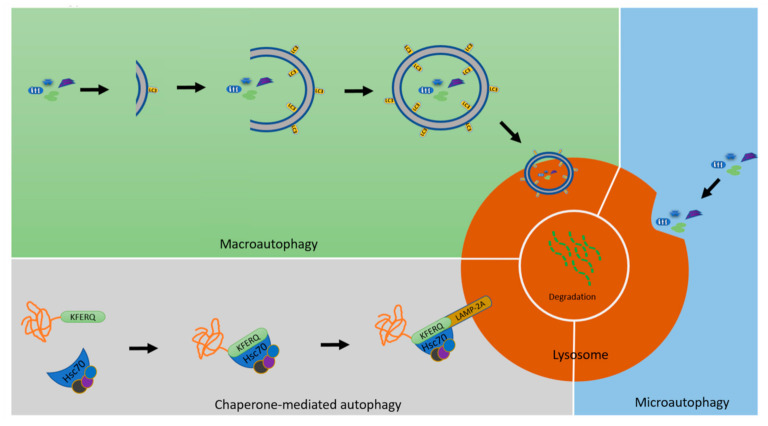
Autophagy Introduction: Autophagy is one of the two main cellular degradative systems along with the ubiquitin proteasome system. The word autophagy (meaning self-eating in Greek) was first coined by Christian de Duve in 1963. During this process, the cytoplasmic components (cargo) including organelles and macromolecules are delivered to lysosomes, where the collection of different acidic hydrolases cause their degradation. This degradation enables the cells to recycle building blocks as well as serves as temporary energy source under conditions like starvation. Based on the route of cargo delivery to lysosomes, the autophagy can be divided into three main types: macroautophagy, microautophagy and chaperon-mediated autophagy. Macroautophagy is the major type of autophagy, during which a cytoplasmic portion containing proteins and organelles is engulfed by autophagosome, which is then fused with lysosome to degrade the engulfed cytoplasmic materials by lysosomal enzymes. In microautophagy, a portion of the cytoplasm is directly delivered to the lysosomes for degradation by lysosomal membrane invagination. Contrary to above two, the chaperon-mediated autophagy specifically degrades proteins containing a specific motif (KFERQ). This involves the formation of a complex between chaperons (Hsc70) and target proteins and then subsequent delivery to lysosome via binding to lysosomal associated membrane protein 2A (LAMP2A).

**Figure 2 cells-11-00038-f002:**
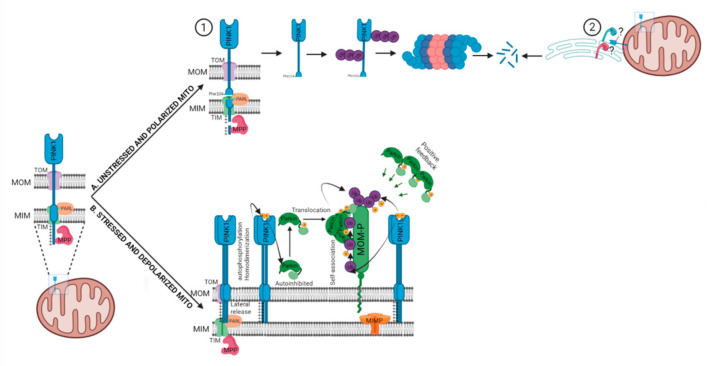
Mitochondrial stress leads to accumulation of PINK1 and PARKIN on mitochondria. Newly translated mitochondrial kinase PINK1 has two fates depending on mitochondrial status. (A) Under unstressed and polarized conditions, the full-length PINK1 containing positively charged mitochondria target sequence (MTS, 1-34 amino acids) at its N-terminus is imported into mitochondria with the help of transporters of the outer and inner membranes (TOM and TIM). The MTS is cleaved off by mitochondrial processing peptidase (MPP) followed by cleavage in the transmembrane domain (TMD) of PINK1 between Ala103 and Phe104, by mitochondrial inner membrane protease PARL. The sequential cleavages generate a shorter PINK1 with the N-terminal Phe104. Two degradation possibilities are proposed for the cleaved PINK1 (1) widely recognized rapid degradation by the proteasome via the N-end rule ubiquitination machinery (2) the second possibility that ubiquitination and degradation of cleaved PINK1 occur at mitochondrial-endoplasmic reticulum (ER) interface by components of the ER-associated degradation pathway, such as the E3 ligases GP78 and HRD1. (B) In contrast to (A), under stressed and depolarized conditions, the import of full-length PINK1 is arrested and it forms a complex with TOM, most likely as a dimer. The PINK1 dimerization is proposed to facilitate PINK1’s autophosphorylation and activation. The mitochondrial outer membrane (MOM) stabilizes full-length PINK1 which phosphorylates pre-existing ubiquitins (Ub) or/and autoinhibited PARKIN. There appears to be no particular order in which PINK1 would phosphorylate Ub or PARKIN first. In both cases, PARKIN translocates to mitochondria and self-associates. The activated PARKIN starts to conjugate ubiquitin to MOM proteins, which are then phosphorylated by PINK1. This forms a positive feedback loop that amplifies the initial signal, resulting in extensive PARKIN recruitment and ubiquitination of MOM proteins. (MOMP and MIMP: mitochondrial outer/inner membrane protein).

**Figure 3 cells-11-00038-f003:**
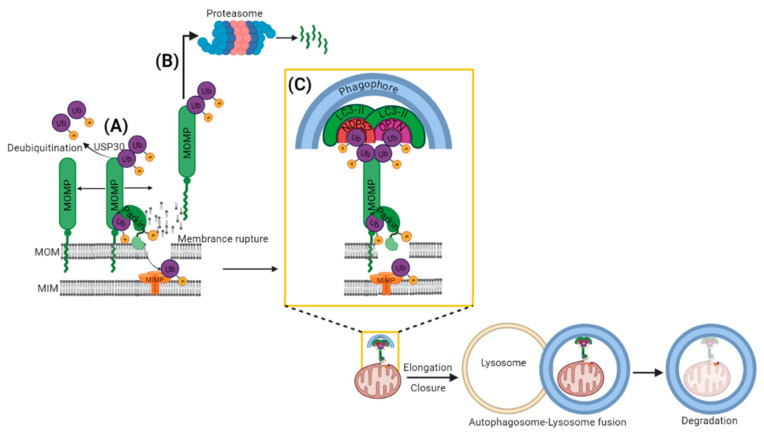
Three fates of PARKIN ubiquitinated mitochondrial proteins. The PARKIN ubiquitinated mitochondrial proteins could have different fates (**A**). The ubiquitinated proteins could undergo deubiquitination by deubiquitinases (DUBs), such as mitochondrially localized ubiquitin carboxyl-terminal hydrolase 30 (USP30). The deubiquitination would remove the polyubiquitin chain, conjugated by PARKIN (shown in Figure 2), and would restore its past status. (**B**) The ubiquitinated proteins could undergo extraction and degradation by the ubiquitin–proteasome system. This leads to the rupture of MOM, which exposes the inner membrane proteins followed by their ubiquitination. (**C**) As the third possibility, the PARKIN-conjugated polyubiquitin chains could initiate the delivery of mitochondria to autophagosomes via the recruitment of the autophagy cargo receptors such as Optineurin (OPTN) and NDP52. The autophagy receptors bind at one end polyubiquitin chains located on mitochondria, and, on another end, they bind LC3 protein, located on autophagosomal membranes (Phagophore). Thus, the autophagy receptors facilitate the delivery of mitochondria to the autophagosome for their further degradation in lysosomes after autophagosomal fusion with lysosomes. (MOMP and MIMP: mitochondrial outer/inner membrane protein).

**Figure 4 cells-11-00038-f004:**
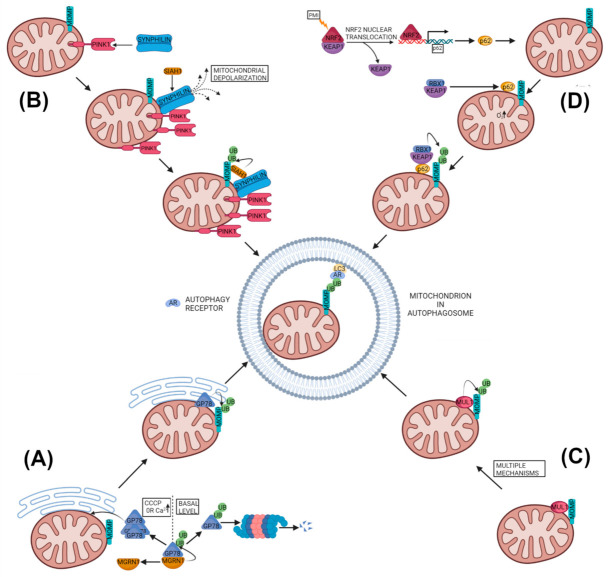
PARKIN-independent but ubiquitin-dependent mitophagies: (**A**) Glycoprotein 78 mitophagy: Glycoprotein 78 (GP78) is an endoplasmic reticulum (ER) membrane–anchored ubiquitin ligase (E3) that is localized to mitochondria-associated ER domains. At the basal level, G78 is polyubiquitinated by the cytosolic E3 ubiquitin ligase mahogunin RING finger 1 (MGRN1) that promotes the degradation of GP78 and maintains low levels of GP78. Whereas, mitochondrial stresses such as CCCP or higher cytosolic Ca^2+^ lead to GP78 accumulation at the mitochondria-associated ER domain triggering PARKIN-independent but ubiquitin-dependent mitophagy. (**B**) PINK1-SYNPHILIN1-SIAH1 mitophagy: The protein SYNPHILIN1 localizes to mitochondria with the help of the mitochondrial kinase PINK1, since SYNPHILIN1 has the affinity towards full length as well as the cleaved form of PINK1. Mitochondrial localization of SYNPHILIN1 causes mitochondrial depolarization, leading to the stabilization of full-length PINK1 onto MOM which further attracts SYNPHILIN1 to the mitochondria. Parallel to PINK1, SYNPHILIN1 also binds to E3 ubiquitin ligase seven in absentia homolog 1 (SIAH1), thus recruiting it to mitochondria. SIAH1 at mitochondria polyubiquitinates MOMP, which leads to recruitment of autophagosomes to mitochondria via the autophagy receptor, and LC3 for mitophagy. (MOMP: mitochondrial outer membrane protein) (**C**) MUL1 based mitophagy: The mitochondrial E3 ubiquitin ligase 1 (MUL1) induces ubiquitin-dependent but PARKIN-independent mitophagy by multiple proposed mechanisms, which are poorly understood and lack consensus. (**D**) p62/SQSTM based mitophagy: p62/SQSTM based mitophagy was observed with the Keap1-Nrf2 PPI inhibitor HB229 (*PMI*). PMI inhibited Keap1-Nrf2 interaction and led to the nuclear translocation of Nrf2. Nuclear translocation of Nrf2 upregulated the expression of p62, which accumulated in mitochondria. Mitochondrial p62 could induce mitophagy either by increasing mitochondrial superoxide or by anchoring the Keap1–RBX1 complex on mitochondria or both. The Keap1–RBX1 complex, being the E3 ubiquitin ligase, polyubiquitinates MOMP, which leads to recruitment of autophagosomes to mitochondria via autophagy receptor and LC3 for mitophagy. (MOMP: mitochondrial outer membrane protein).

**Figure 5 cells-11-00038-f005:**
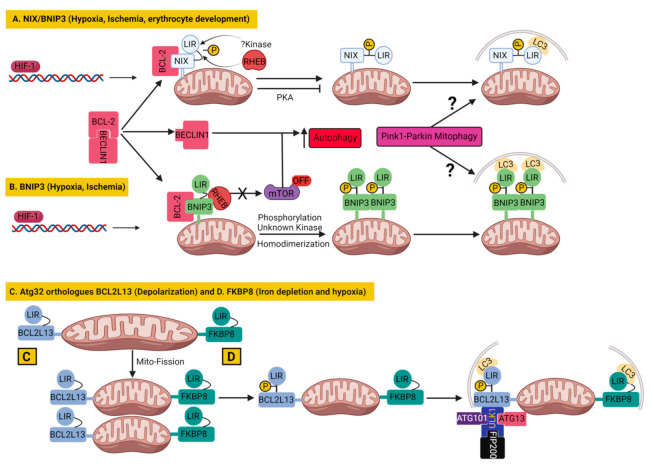
Apoptosis-related proteins as Mitophagy receptors: (**A**) NIX/BNIP3L mediated mitophagy: Apoptotic death-inducing mitochondrial protein NIX/BNIP3L mediates mitophagy under ischemia and erythrocyte development. The promoter region of *NIX/BNIP3L* gene contains a binding site for hypoxia-inducible factor 1 (HIF-1), which upregulates its expression under hypoxic conditions. The mitophagy activity of NIX/BNIP3L is regulated by its phosphorylation by small GTPase RHEB and an unknown kinase close to its LIR motif. This phosphorylation promotes interaction between Nix/BNIP3L and LC3, leading to autophagosomal recruitment to mitochondria for mitophagy. In contrast, phosphorylation of Nix/BNIP3L by PKA inhibits mitophagy. NIX/BNIP3 activates general autophagy by sequestering BCL-2 from the BCL-2 and BECLIN1 complex. (**B**) BNIP3-mediated mitophagy: Like NIX/BNIP3L, BNIP3 also induces mitophagy under hypoxic and ischemic conditions. The promoter region of the *BNIP3* gene also contains the binding site for hypoxia-inducible factor 1 (HIF-1), which upregulates its expression under hypoxic conditions. BNIP3 also activates general autophagy by sequestering BCL-2 from the BCL-2 and BECLIN1 complex. Additionally, BNIP3 also activates autophagy by sequestering RHEB, the activator of the mTOR that leaves mTOR inactivated. The mitophagy activity of BNIP3 is regulated by its homodimerization and its double phosphorylation nears its LIR domain by an unknown kinase. Both modifications promote interaction between BNIP3 and LC3, leading to autophagosomal recruitment to mitochondria for mitophagy. (**C**,**D**) ATG32-orthologue-mediated mitophagy: BCL2L13 (**C**) and FKBP8 (**D**) are mammalian orthologues of yeast mitophagy receptor ATG32. BCL2L13 and FKBP8 both contain LIR motifs, and both induce DRP1-independent mitochondrial fission. The phosphorylation of BCL2L13 promotes its mitophagy activity. BCL2L13 also recruits the ULK1 complex to process mitophagy post or parallel to LC3B recruitment to mitochondria. FKBP8 mediates mitophagy by specifically interacting with LC3A that recruits the autophagosome to mitochondria for mitophagy.

**Figure 6 cells-11-00038-f006:**
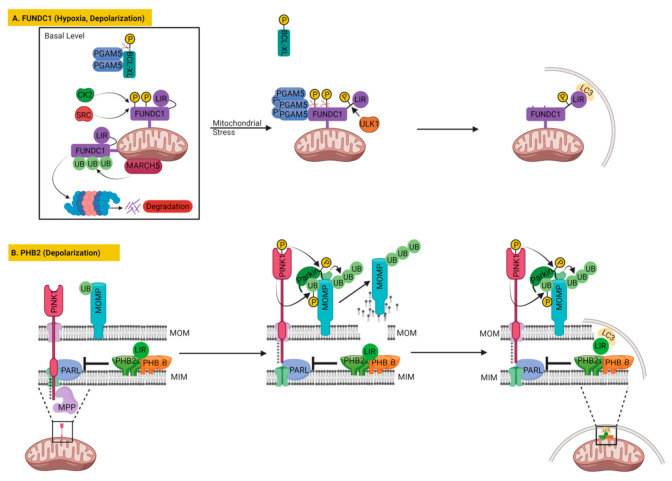
Other key mitophagy receptors: (**A**) FUNDC1 mediated mitophagy: The mitochondrial protein FUNDC1 is a key mitophagy receptor under depolarization and hypoxia. At the basal level, FUNDC1 activity is regulated by two posttranslational modifications. First, by phosphorylation via the protein kinases non-receptor tyrosine kinase (SRC) and casein kinase 2 (CK2), which prevents the interaction between FUNDC1 and LC3 through the LIR motif. The dephosphorylation of phosphorylated FUNDC1 by its dephosphorylase PGAM5 does not occur under basal conditions, due to the binding of PGAM5 to BCL-XL. Secondly, the FUNDC1 activity is also regulated by ubiquitination via mitochondrial E3 ubiquitin ligase MARCH5, which leads to its proteasomal degradation. However, under mitochondrial stress conditions, PGAM5 dissociates from BCL-XL, followed by its multimerization and dephosphorylation of FUNDC1. Concomitantly, the FUNDC1 is phosphorylated by ULK1. These post-translational modifications promote the interaction of FUNDC1 with LC3, which results in mitophagy. (**B**) Prohibitin 2 (PHB2)-mediated mitophagy: PHB2 is a novel mitochondrial inner membrane mitophagy receptor and it is part of the prohibitin complex. Unlike other outer membrane mitophagy receptors, PHB2 accelerates mitophagy by interacting with LC3 via its LIR after mitochondrial outer membrane (MOM) rupture. According to the recently proposed mechanism, PHB2 promotes the stabilization of PINK1 onto MOM by inhibiting PARL, which cleaves PINK1 under basal conditions (see Figure 2). Stabilization of PINK1 onto MOM promotes PARKIN translocation and ubiquitination of MOMPs. Ubiquitination of MOMPs results in MOMPs degradation and MOM rupture, which exposes the LIR motif of PHB2. The exposure of the LIR motif enables PHB2 to interact with LC3, which recruits autophagosomes to mitochondria for mitophagy. (MOMPs: mitochondrial outer membrane proteins).

**Figure 7 cells-11-00038-f007:**
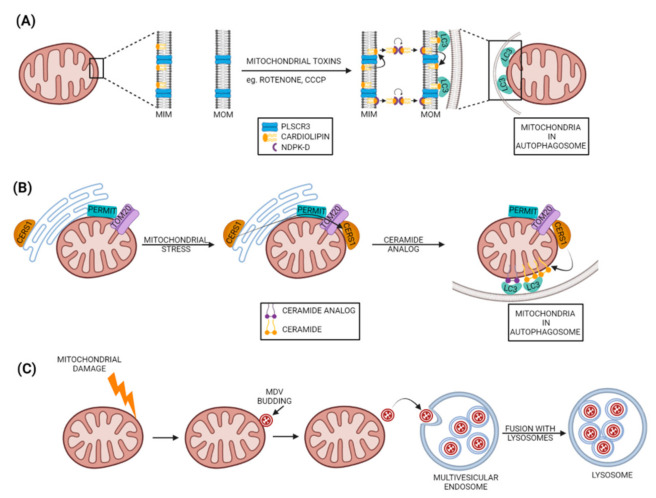
Lipids as mitophagy receptors and micromitophagy: (**A**) Cardiolipin-mediated mitophagy: Cardiolipin (CL) is a predominantly mitochondrial inner membrane (MIM) lipid. The mitochondrial toxins promote the externalization of CL which is mediated by mitochondrial phospholipid scramblase-3 (PLSCR3) and by intermembrane space protein complex of mitochondrial nucleoside diphosphate kinase D (NDPK-D). PLSCR3 flips the CL from one membrane side to other, whereas NDPK-D transports CL across the intermembrane space. Once at the MOM, the CL directly recruits LC3 to mitochondria by binding to the N-terminal helix of LC3 independently of mitochondria depolarization to induce mitophagy. (**B**) Ceramide-mediated mitophagy: The accumulation of ceramide or ceramide analog at mitochondria promotes mitophagy, since they can directly interact with LC3 in the autophagosome. According to the recently proposed mechanism, mitochondrial stress signaling promotes the trafficking of newly translated CerS1 (responsible for synthesis of ceramides) from the ER surface to MOM through mitochondria-associated membranes (MAMs) by a protein that mediates ER –mitochondria Trafficking p17/PERMIT by interacting with TOM20. The mitochondrially translocated CerS1 starts C18-ceramide generation, which induces the LC3B-II-mediated targeting of autophagosomes to mitochondria for lethal mitophagy. Similar mitophagy takes place with a ceramide analog, which accumulates on mitochondria. (**C**) Micromitophagy: Externally or internally damaged parts of mitochondria are packed in mitochondria-derived vesicles (MDV) that bud off from the mitochondria. The MDV is then delivered to lysosomes via multivesicular endosomes. Degradation of MDVs takes place in the lumen of lysosomes.

## Data Availability

Not applicable.

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
