# Peer review of "Molecular Mechanisms and Regulation of Mammalian Mitophagy"

_cells, 2021, doi:10.3390/cells11010038_

Round 1

Reviewer 1 Report

General comment:

In the manuscript entitled " An overview of mammalian mitophagy: mechanistic details and implication in human health and diseases” Choubey and colleagues present a review of the literature about mitophagy in mammals.

After a short introduction, the manuscript is organized in four parts presenting the main pathways involved in macromitophagy (parts 1-3) or micromitophagy (part 4), before two conclusive paragraphs. The review is interesting, informative and relevant for the field. It covers a wide range of aspects about mitophagy and many references are cited including recently published data.

Specific comments:

The major remark concerns the discrepancy between the title of the review and its content.

While the “mechanistic details” mentioned in the title are exhaustive and very well covered, the “implication in human health and diseases” is far less described and somehow disappointing. The manuscript would benefit of more information on the physiological aspects and the pathologies associated to mitophagy. Mitophagy is involved in several biological processes such as development, cell differentiation, immunity or longevity which are not mentioned in the review. Moreover, the pathological aspects linking mitophagy to viral infection, genetic diseases or neurodegenerative disorders are not enough described. In this regard, either the title of the review should be modified or the second part strongly reinforced in the text and accompanied with a table or a figure.

Except this main remark the manuscript is well-structured and easy to read, and the other remarks mainly concern minor modifications in the figures and the text.

The review is well illustrated but Figures 1, 2 and 3 are not easy to read at a glance and their understanding would be easier if they were homogenized with Figures 4 to 6 (reading from left to right) and increasing size of the protein names in Figure 3.

The part on micromitophagy is very interesting and adding a corresponding figure could be helpful for the reader and illustrate the existence of a microendosomal mitophagy.

The last two paragraphs of the review named “Mitophagy and Mitochondrial biogenesis could be linked” and “Future direction:” are not numbered. Are they individual chapters or subparts of the conclusion?

Other minor corrections:

Page 2 In introduction : (See autophagy Box). No corresponding box in the manuscript.

Page 21 part 3.A: Cardi-olipin

Page 22 Figure 6 : CERMIDE

Page 22 last line : The BiorXiv reference should be formatted and added in the reference list.

Author Response

Response to the Reviewer 1:

Authors Response: First of all, authors would like thank reviewer for the positive and constructive comments. We have attempted to provide answers to the all your comments in a point-by-point manner. Please find our answers in Italicized red fonts.  

General comment:

In the manuscript entitled " An overview of mammalian mitophagy: mechanistic details and implication in human health and diseases” Choubey and colleagues present a review of the literature about mitophagy in mammals.

After a short introduction, the manuscript is organized in four parts presenting the main pathways involved in macromitophagy (parts 1-3) or micromitophagy (part 4), before two conclusive paragraphs. The review is interesting, informative and relevant for the field. It covers a wide range of aspects about mitophagy and many references are cited including recently published data.

Specific comments:

The major remark concerns the discrepancy between the title of the review and its content.

While the “mechanistic details” mentioned in the title are exhaustive and very well covered, the “implication in human health and diseases” is far less described and somehow disappointing. The manuscript would benefit of more information on the physiological aspects and the pathologies associated to mitophagy. Mitophagy is involved in several biological processes such as development, cell differentiation, immunity or longevity which are not mentioned in the review. Moreover, the pathological aspects linking mitophagy to viral infection, genetic diseases or neurodegenerative disorders are not enough described. In this regard, either the title of the review should be modified or the second part strongly reinforced in the text and accompanied with a table or a figure.

Authors Response: We agree with the reviewer that manuscript would have been more informative if more mitophagy associated physiological aspects and pathologies were discussed. However, the time given for revision (5 days) is too short for such addition. Therefore, as per suggestion of reviewer we are modifying the title of our review. The new title is “Molecular Mechanisms and Regulation of Mammalian Mitophagy”.

Except this main remark the manuscript is well-structured and easy to read, and the other remarks mainly concern minor modifications in the figures and the text.

The review is well illustrated but Figures 1, 2 and 3 are not easy to read at a glance and their understanding would be easier if they were homogenized with Figures 4 to 6 (reading from left to right) and increasing size of the protein names in Figure 3.

Authors Response: According to the suggestions of the reviewer Figure 1 and 2 (in revised version figure 2 and 3) are reorganized from left to right to homogenize with Figure 4 to 6 (in revised version figure 5 to 7). But reorganizing multiple mitophagy pathways from left to right in Figure 3 (in revised version figure 4) couldn’t be done due to overcrowding. However, according to the suggestion of reviewer the font size of protein names in the figure 3 (in revised version figure 4) were increased from 10 to 12.      

The part on micromitophagy is very interesting and adding a corresponding figure could be helpful for the reader and illustrate the existence of a microendosomal mitophagy.

Authors Response: According to the suggestion of the reviewer an illustration of micromitophagy is added to figure 7 (C) along with corresponding addition in the figure 7 legend.

The last two paragraphs of the review named “Mitophagy and Mitochondrial biogenesis could be linked” and “Future direction:” are not numbered. Are they individual chapters or subparts of the conclusion?

Authors Response: Thank you very much for pointing this out. They are individual chapters and to mark that in the revised version bullets are added to following parts ‘’Introduction’’, ‘’Mitophagy and Mitobiogenesis could be llinked’’ and ‘’Future Direction’’.

Other minor corrections:

Page 2 In introduction : (See autophagy Box). No corresponding box in the manuscript.

Authors Response: Thank you very much for pointing this out. To curb ambiguity, in the revised manuscript ‘’autophagy box’’ is replaced with ‘’Figure 1’’ in introduction.

Page 21 part 3.A: Cardi-olipin

Authors Response: Thank you very much for pointing this out. This is because of formatting by the journal before sending it for review. In the manuscript version sent to me for correction, the word cardiolipin appeared with hyphen twice when there was change of line.   

Page 22 Figure 6 : CERMIDE

Authors Response: The spelling of Ceramide is corrected in Figure 6B (revised version Figure 7B).

Page 22 last line : The BiorXiv reference should be formatted and added in the reference list.

Authors Response: The BiorXiv reference is formatted and added in the reference list (in revised manuscript, Reference number 241).

Yours sincerely,

Vinay Choubey, PhD

Department of Pharmacology

University of Tartu

Ravila 19, 51014 Tartu

ESTONIA

Reviewer 2 Report

In this review Choubey et al provide an extensive overview of mammalian mitophagy and its importance in human health and diseases.

The manuscript is well written and organized. The figures are excellent.

As a suggestion, it could be interesting to include a discussion about the role of mitophagy in cellular senescence, the coordinated actions of mitophagy and UPRmt and therapeutic strategies targeting mitophagy.

Author Response

Reviewer 2:

Authors Response: First of all, authors would like thank reviewer for the positive and constructive comments. We have attempted to provide answer to the your comment in a point-by-point manner. Please find our answers in Italicized red fonts.  

In this review Choubey et al provide an extensive overview of mammalian mitophagy and its importance in human health and diseases.

The manuscript is well written and organized. The figures are excellent.

As a suggestion, it could be interesting to include a discussion about the role of mitophagy in cellular senescence, the coordinated actions of mitophagy and UPRmt and therapeutic strategies targeting mitophagy.

Authors Response: We agree with the reviewer that inclusion of the above mentioned topics would have surely enriched the manuscript. However, the time given for revision (5 days) is too short for such addition. Therefore, we are modifying the title of our review. The new title is “Molecular Mechanisms and Regulation of Mammalian Mitophagy”.

Yours sincerely,

Vinay Choubey, PhD

Department of Pharmacology

University of Tartu

Ravila 19, 51014 Tartu

ESTONIA
